# Modelling the development and decay of cryoconite holes in Northwest Greenland

Yukihiko Onuma[1], Koji Fujita[2], Nozomu Takeuchi[3], Masashi Niwano[4] and Teruo Aoki[5]

[1]Earth Observation Research Center (EORC), Japan Aerospace Exploration Agency (JAXA), Tsukuba, 305-8505, Japan
[2]Graduate School of Environmental Studies, Nagoya University, Nagoya 464-8601, Japan
[3]Graduate School of Science, Chiba University, Chiba 263-8522, Japan
[4]Meteorological Research Institute, Japan Meteorological Agency, Tsukuba 305-0052, Japan
[5]National Institute of Polar Research, Tokyo 190-8518, Japan

*Correspondence to*: Yukihiko Onuma[1] and Koji Fujita[2] ([1]onuma.yukihiko@jaxa.jp, [1]yonuma613@gmail.com and [2]cozy@nagoya-u.jp)

**Abstract.** Cryoconite holes (CHs) are water-filled cylindrical holes with cryoconite (dark-coloured sediment) deposited at their bottoms, forming on ablating ice surfaces of glaciers and ice sheets worldwide. Because the collapse of CHs may disperse cryoconite on the ice surface, thereby decreasing the ice surface albedo, accurate simulation of the temporal changes in CH
depth is essential for understanding ice surface melt. We established a novel model that simulates the temporal changes in CH depth using heat budgets calculated independently at the ice surface and CH bottom based on hole-shape geometry. We evaluated the model with in situ observations of the CH depths on the Qaanaaq ice cap in Northwest Greenland during the 2012, 2014, and 2017 melt seasons. The model reproduced well the observed depth changes and timing of CH collapse. Although earlier models have shown that CH depth tends to be deeper when downward shortwave radiation is intense, our
sensitivity tests suggest that deeper CH tends to form when the diffuse component of downward shortwave radiation is dominant, whereas CHs tend to be shallower when the direct component is dominant. In addition, the total heat flux to the CH bottom is dominated by shortwave radiation transmitted through ice rather than that directly from the CH mouths when the CH is deeper than 0.01 m. Because the shortwave radiation transmitted through ice can reach the CH bottom regardless of CH diameter, CH depth is unlikely to be correlated with CH diameter. The relationship is consistent with previous observational
studies. Furthermore, the simulations highlighted that the difference in albedo between ice surface and CH bottom was a key factor for reproducing the timing of CH collapse. It implies that lower ice surface albedo could induce CH collapse and thus cause further lowering of the albedo. Heat component analysis suggests that CH depth is governed by the balance between the intensity of the diffuse component of downward shortwave radiation and the turbulent heat transfer. Therefore, these meteorological conditions may be important factors contributing to the recent surface darkening of the Greenland ice sheet
and other glaciers via the redistribution of CHs.

## 1. Introduction

Cryoconite holes (CHs), cylindrical water-filled holes formed on ablating ice surfaces of glaciers and ice sheets worldwide, influence the ablation process of glacial ice and function as the habitats of glacial microbes (e.g., Hodson et al., 2010a; Cook et al., 2016; Uetake et al., 2016; Zawierucha et al., 2021). In CHs, dark-coloured sediment, referred to as cryoconite, is deposited at the bottoms; it consists mostly of spherical aggregates (cryoconite granules) formed by the entanglement of filamentous cyanobacteria with mineral and organic particles (Hodson et al., 2010b; Langford et al., 2010; Takeuchi et al., 2010, 2014; Wientjes et al., 2011; Uetake et al., 2019). Because such dark-coloured sediment on bare ice absorbs solar radiation more effectively than the surrounding clear ice, CH formation occurs often.

The development and decay of CHs depend on the meteorological conditions near the ice surface. CHs are likely to form well under sunny conditions (McIntyre, 1984; Fountain et al., 2008; Jepsen et al., 2010; Cook et al., 2016; Takeuchi et al., 2018). Although many studies have investigated which meteorological conditions are favourable for the deepening (development) of CHs, there is limited information on their shallowing (decay) dynamics. Takeuchi et al. (2018) reported that CHs collapse under cloudy and windy conditions in summer in Northwest Greenland. They suggested that CHs collapse when the ice surface surrounding the CHs experiences enhanced melt caused by turbulent heat flux exchange. CH collapse events have also been reported in Svalbard and Southwest Greenland (Hodson et al., 2007, 2008; Stibal et al., 2008; Irvine-Fynn et al., 2011; Chandler et al., 2015), while it has been indicated that a higher melt rate at the ice surface induces CH collapse (Hodson et al., 2007).

The development and decay of CHs modulate the ice surface albedo. On the one hand, the development of CHs increases the area-averaged ice surface albedo compared to the case where the surface is uniformly covered with cryoconite granules, because cryoconite granules in holes are shielded from sunlight by the hole walls (Bøggild et al., 2010; Takeuchi et al., 2014; Chandler et al., 2015). On the other hand, the collapse of CHs reduces the area-average surface albedo by approximately 0.1–0.2 due to the redistributed cryoconite on the ice surface (Irvine-Fynn et al., 2011; Takeuchi et al., 2018). Topologically heterogeneous ice surfaces of the Greenland ice sheet (GrIS) can be classified into four types: clean bare ice surfaces, dirty bare ice surfaces, surfaces with CHs, and meltwater streams (Irvine-Fynn and Edwards, 2014; Chandler et al., 2015; Holland et al., 2019; Tedstone et al., 2020). Bare ice surfaces and surfaces with CHs are estimated to account for approximately 80–90 and 5–15 %, respectively, of the southwestern GrIS ablation area (Chandler et al., 2015). Therefore, both the development and decay of CHs are important for the surface albedo dynamics of the GrIS.

Numerical models have been used for reproducing the temporal changes in CH depth; however, only few models consider the decay of CHs. Gribbon (1979) proposed a conceptual model based on the Beer–Lambert law to simulate the development of CHs. By using solar radiation and CH depth, this model calculates the shortwave radiation reaching the cryoconite at the CH bottoms using the absorption coefficient of ice. Although several studies have proposed numerical models following Gribbon (Podgorny and Grenfell, 1996; Jepsen et al., 2010; Cook, 2012), these models focus only on the development of CHs.

Therefore, to reproduce CH decay processes, such as CH collapse events, a model that simulates melt not only at the CH bottom, but also at the ice surface surrounding the CHs is necessary.

In this study, we established a numerical model to simulate the temporal changes in CH depth, accounting for both the development and decay of CHs, based on a concept different from those in previous studies [e.g., Gribbon (1979)]. We evaluated the model using observed temporal changes in CH depth from the Qaanaaq ice cap in Northwest Greenland. We further performed numerical model sensitivity analyses to reveal how the CH depth is controlled by the meteorological variables, hole geometry, albedos of the ice surface surrounding the CHs and the cryoconite therein, and optical parameters of

ice. Finally, we assessed the meteorological factors controlling the CH depth changes based on heat component analysis.

## 2. Cryoconite hole model (CryHo)

The cryoconite hole model (hereafter referred to as CryHo) developed in this study calculates the heat balances and resultant melt rates at the ice surface and CH bottom independently (Fig. 1). All variables and constants described in the following equations are listed in Table 1.

**2.1. Heat balance at the ice surface**

The heat balance at the ice surface ($Q_i$, W m$^{-2}$) is described as follows:

$$Q_i = (1 - \alpha_i)R_S + R_{Lni} + H_{Si} + H_{Li}, \tag{1}$$

$$R_{Lni} = \varepsilon R_L - \varepsilon \sigma T_i^4, \tag{2}$$

where $\alpha_i$ is the ice surface albedo (dimensionless), $R_S$ is the downward shortwave radiation (W m$^{-2}$), $R_{Lni}$ is the net longwave

radiation (W m$^{-2}$), $H_{Si}$ is the sensible heat flux (W m$^{-2}$), $H_{Li}$ is the latent heat flux (W m$^{-2}$), $\varepsilon$ is the emissivity of the snow/ice surface, which is assumed to be 1.0 (dimensionless), $R_L$ is the downward longwave radiation (W m$^{-2}$), $\sigma$ is the Stefan–Boltzmann constant ($5.67 \times 10^{-8}$ W m$^{-2}$ K$^{-4}$), and $T_i$ is the surface temperature (K). Subscript $i$ used in the variables refers to the ice surface. All the downward components have positive signs. Note that $H_{Li}$ is restricted to the latent heat of evaporation; it does not include the latent heat of melting. Heat conduction from/toward the glacier ice and heat supplied by rainwater are

assumed to be negligible. The turbulent heat fluxes (i.e., $H_{Si}$ and $H_{Li}$) were estimated using the following bulk formulations:

$$H_{Si} = c_P \rho_a CU(T_a - T_i), \tag{3}$$

$$H_{Li} = l_E \rho_a CU\big(h_r q(T_a) - q(T_i)\big), \tag{4}$$

where $c_P$ is the specific heat of air (1,006 J K$^{-1}$ kg$^{-1}$), $\rho_a$ is the air density (kg m$^{-3}$), $C$ is the bulk coefficient for the snow/ice surface (0.0025, Kondo, 1994), $U$ is the wind speed (m s$^{-1}$), $T_a$ is the air temperature (K), $l_E$ is the latent heat of water evaporation (2.50 × 10$^6$ J kg$^{-1}$), $h_r$ is the relative humidity (dimensionless), and $q(T)$ is the saturated specific humidity (kg kg$^{-1}$). The excess energy ($Q_i > 0$) is used to melt the ice at the surface, once the surface temperature reaches the melting point (273.15 K), and its hourly amount is obtained as follows:

$$Q_{Mi} = \max[0, Q_i], \tag{5}$$

$$M_i = \frac{t_h Q_{Mi}}{l_M \rho_i},$$

where $Q_{Mi}$ is the excess heat energy for ice melt (W m$^{-2}$), $M_i$ is the melting ice thickness (m h$^{-1}$), $t_h$ is the length of an hour in seconds (3,600 s), $l_M$ is the latent heat of ice melt (3.33 × 10$^5$ J kg$^{-1}$), and $\rho_i$ is the ice density assumed equal to 900 kg m$^{-3}$.

## 2.2. Heat balance at the cryoconite hole bottom

The heat balance and melt rate at the CH bottom (denoted by subscript $c$) are defined as follows:

$$Q_c = (1 - \alpha_c)(R_{Sdc} + R_{Sfc} + R_{Stdc} + R_{Stfc}) + R_{Lnc}, \tag{6}$$

$$Q_{Mc} = \max[0, Q_c], \tag{7}$$

$$M_c = \frac{t_h Q_{Mc}}{l_M \rho_i},$$

where $Q_c$ is the heat balance at the bottom (W m$^{-2}$); $\alpha_c$ is the albedo (dimensionless); $R_{Sdc}$ and $R_{Sfc}$ are the direct and diffuse components of shortwave radiation (W m$^{-2}$); $R_{Stdc}$ and $R_{Stfc}$ are the direct and diffuse components of shortwave radiation transmitted through ice (W m$^{-2}$); $R_{Lnc}$ is the net longwave radiation (W m$^{-2}$); $Q_{Mc}$ is the excess heat energy for ice melt (W m$^{-2}$); and $M_c$ is the melting ice thickness (m h$^{-1}$). Because we assume that the CH is partially filled with water, the turbulent heat fluxes at the bottom of the CH (i.e., $H_{Sc}$ and $H_{Lc}$) are assumed to be zero in Eq. (6). The change in CH depth ($D_t$, m) at a given time ($t$) is defined as follows:

$$D_t = D_{t-1} + (M_c - M_i), \tag{8}$$

where $D_{t-1}$ is the CH depth at one time step before (m). If the melt rate at the CH bottom is greater than that at the ice surface ($M_c > M_i$), the CH depth deepens, and vice versa. The initial depth $D_0$ at $t = 0$ in CryHo is a prescribed constant initial condition. For simplicity, we assume the heat balance at the centre of the CH bottom can represent that of the entire bottom.

### 2.3. Geometry effect on the radiation components

In earlier CH models (Gribbon, 1979; Jepsen et al., 2010; Cook, 2012), downward shortwave radiation is reduced exponentially with increasing CH depth. The novel aspect of our model is that it considers not only the CH depth, but also the CH shape geometry. The direct and diffuse components of downward shortwave radiation and downward longwave radiation are also considered independently. Regarding downward shortwave radiation, we first separate the observed downward shortwave radiation into direct and diffuse components ($R_{Sd}$ and $R_{Sf}$, W m$^{-2}$) as follows:


$$R_{Sd} = (1 - r_{dif})R_S, \tag{9}$$
$$R_{Sf} = r_{dif}R_S, \tag{10}$$

where $r_{dif}$ is the diffuse ratio of downward shortwave radiation (dimensionless; 0–1), which is calculated following Goudriaan (1977):

$$r_{dif} = r_{ze} + (1 - r_{ze})r_{cld}, \tag{11}$$


where $r_{ze}$ is the ratio based on the zenith angle (dimensionless; 0–1); and $r_{cld}$ is the ratio based on cloudiness (dimensionless; 0–1). Calculation of $r_{ze}$ was performed following Goudriaan (1977), and calculation of $r_{cld}$ was performed following Broeke et al. (2004), and Niwano et al. (2015):

$$r_{ze} = \frac{0.0604}{\max[0.01, (\cos\theta_z - 0.0223)]} + 0.0683, \tag{12}$$
$$r_{cld} = 1 - \frac{R_{Lni}}{[1363.2 - 5.4T_a]}, \tag{13}$$


where $\theta_z$ is the solar zenith angle (radian).

The CH geometry is defined as the zenith angle of the edge from the centre of the CH bottom ($\theta_c$, rad) as follows:

$$\theta_c = \tan^{-1}\left(\frac{\phi}{2D_{t-1}}\right), \tag{14}$$

where $\phi$ is the diameter of CH (m).

Compared with the solar zenith angle, the direct component of shortwave radiation from the sky looking up from the CH bottom ($R_{Sdc}$; W m$^{-2}$) is calculated as follows:

$$R_{Sdc} = R_{Sd} \quad [if \ \theta_z \le \theta_c], \tag{15}$$
$$R_{Sdc} = 0 \quad\quad [if \ \theta_z > \theta_c].$$

Meanwhile, the diffuse component of shortwave radiation from the sky looking up from the CH bottom ($R_{Sfc}$, W m$^{-2}$) is obtained as follows:

$$R_{Sfc} = (1 - \cos^2 \theta_c)R_{Sf}. \tag{16}$$

Shortwave radiation components transmitted through ice are described in the following section.

Longwave radiation from the sky looking up from the CH bottom ($R_{Lc}$; W m$^{-2}$) can be obtained in a manner similar to $R_{Sfc}$ as follows:

$$R_{Lc} = (1 - \cos^2 \theta_c)\varepsilon R_L. \tag{17}$$

Longwave radiation from the CH wall ($R_{Lw}$; W m$^{-2}$) is subsequently described as follows:


$$R_{Lw} = \cos^2 \theta_c \ \varepsilon\sigma T_c{}^4. \tag{18}$$

Here, we assume that the CH wall or bottom temperatures ($T_c$; K) are equal to the melting point. Because the CryHo does not calculate water level in the CH, the longwave radiation emitted from the CH wall is calculated using snow/ice surface emissivity, which is similar to the water emissivity in Eq. (18). Regarding longwave radiation emitted from the CH bottom,

the net longwave radiation ($R_{Lnc}$; W m$^{-2}$) can be summarised as follows:

$$R_{Lnc} = R_{Lc} + R_{Lw} - \varepsilon\sigma T_c{}^4 = (1 - \cos^2 \theta_c)R_{Lni}. \tag{19}$$

## 2.4. Shortwave radiation transmitted through ice

Depending on whether direct solar radiation illuminates the CH bottom ($\theta_z \leq \theta_c$) or not ($\theta_z > \theta_c$), the direct and diffuse components of shortwave radiation transmitted through the ice ($R_{Stdc}$ and $R_{Stfc}$, W m$^{-2}$) are described as follows:

$$R_{Stdc} = e^{\frac{-\kappa_d D_{t-1}}{\cos \theta_z}} R_{Sd} \quad [if\ \theta_z > \theta_c], \tag{20}$$
$$R_{Stdc} = 0 \quad\quad\quad\quad [if\ \theta_z \leq \theta_c],$$
$$R_{Stfc} = \cos^2 \theta_c\, e^{-\kappa_f D_{t-1}} R_{Sf}, \tag{21}$$

where $\kappa_d$ and $\kappa_f$ are the broadband asymptotic volume flux extinction coefficients (hereafter broadband flux extinction coefficients) of ice (m$^{-1}$) for the direct and diffuse components, respectively. For the direct component, the path length of the transmitted light is considered, whereas for the diffuse component, it is assumed to be the CH depth.

The broadband flux extinction coefficient for the diffuse component ($\kappa_f$) can be calculated from the spectral flux extinction coefficients. However, this is not a simple integration because the spectral distribution of shortwave radiation transmitted through the ice varies depending on both the solar illumination spectral distribution and the path length in the ice. Thus, we first calculate the insolation spectral distribution in the ice from both the spectral flux extinction coefficients experimentally determined by Cooper et al. (2021) for ice on the GrIS (the term "irradiance attenuation coefficient" was used in their study) and the spectral solar radiation at the ice surface assumed for clear and cloudy skies, computed using a radiative transfer model (Aoki et al., 1999; 2000).

### 2.4.1. Theoretical basis for calculating bare ice broadband flux extinction coefficient

Section 2.4.1. presents an expression for the broadband asymptotic volume flux extinction coefficient, $\kappa$ (m$^{-1}$), from the spectral asymptotic volume flux extinction coefficient $k_e(\lambda)$ (m$^{-1}$) of bare ice, where $\lambda$ is the wavelength (μm). In this study, we simply refer to them as the "broadband (or spectral) flux extinction coefficient." When the bare ice surface is illuminated by the spectral solar radiation $F_s(\lambda)$, the attenuated solar radiation $F_b(\lambda)$ at ice thickness $x$ (m) in bare ice is expressed by $k_e(\lambda)$ as follows:

$$F_b(\lambda) = F_s(\lambda)\exp\left[-k_e(\lambda)x\right]. \tag{22}$$

Using Eq. (22), the spectral flux transmittance ($T(\lambda, x)$; dimensionless) as a function of $x$ is described as follows:

$$T(\lambda, x) = \frac{F_b(\lambda)}{F_s(\lambda)} = \exp\left[-k_e(\lambda)x\right]. \tag{23}$$

The spectrally integrated flux transmittance $\bar{T}(x)$ is calculated from $F_s(\lambda)$ and $T(\lambda, x)$ as follows:


$$\bar{T}(x) = \frac{\int_0^\infty F_s(\lambda)T(\lambda,x)d\lambda}{\int_0^\infty F_s(\lambda)d\lambda}. \tag{24}$$

Using Eq. (24), the broadband flux extinction coefficient ($\overline{k_e}(x)$, m$^{-1}$) is described as follows:

$$\overline{k_e}(x) = \frac{-ln(\bar{T}(x))}{x} = -ln\left(\frac{\int_0^\infty F_s(\lambda)\exp\left[-k_e(\lambda)x\right]d\lambda}{\int_0^\infty F_s(\lambda)d\lambda}\right)/x. \tag{25}$$

### 2.4.2. Parameterization of broadband flux extinction coefficient

Using Eq. (25), we calculate the $\overline{k_e}(x)$ for bare ice in Greenland from the following two datasets for $k_e(\lambda)$ and $F_s(\lambda)$. The first data set $k_e(\lambda)$ is employed based on measurements by Cooper et al. (2021) for the bare ice "Layer A" and "Layer B," which are defined as the ice layers from 12–77 and 53–124 cm depth, respectively. The second dataset $F_s(\lambda)$ is theoretically simulated at $\lambda = 0.2$– 4.0 μm with a spectral resolution of 0.025 μm using a radiative transfer model for the atmosphere– snow/ice system (Aoki et al., 1999; 2000), assuming clear and cloudy sky conditions in the subarctic model atmosphere over

the bare ice surface. As the spectral range of $k_e(\lambda)$ reported by Cooper et al. (2021) is limited around the visible wavelengths ($\lambda = 0.35 - 0.75$μm), with their Table A1 data ($\lambda = 0.35$– 0.60 μm) and the calculated values from their transmittance data ($\lambda = 0.60$– 0.75 μm) being employed in this study, we assume $k_e(\lambda)$ in the other spectral regions $\lambda = 0.2$– 0.35 μm and $\lambda = 0.75$– 4.0 μm by scaling the pure ice spectral absorption coefficient $k_a(\lambda)$ (Warren and Brandt, 2008) to connect to the values of Cooper et al. (2021), as shown in Figure 2. It is inferred that the difference between the scaled $k_e(\lambda)$ (pink curve) and pure

ice $k_a(\lambda)$ (blue curve and red "x" symbols) is attributed to air bubbles in bare ice for $\lambda \geq 0.55$ μm, and impurities and air bubbles for $\lambda < 0.55$ μm from the discussion by Cooper et al. (2021). There are large uncertainties in $k_a(\lambda)$ in $\lambda = 0.35$– 0.75 μm, as shown by the blue curve (Warren and Brandt, 2008) and the red "x" symbols (Cooper et al., 2021) in Figure 2. Regarding the scaled $k_e(\lambda)$ validity in other spectral regions, we did not have verifiable data. In fact, the contribution to $\overline{k_e}(x)$ from $k_e(\lambda)$ in the other spectral regions is very low. The simulated $F_s(\lambda)$ for clear and cloudy skies for the solar zenith

angle $\theta_z = 63.1°$ is shown in Figure S1.

Figure S2 demonstrates $\overline{k_e}(x)$ as a function of $\theta_z$ for $x = 0.1$ m calculated from $k_e(\lambda)$ and $F_s(\lambda)$ described above using Eq. (25). There is almost no difference in $\overline{k_e}(x)$ when we use $k_e(\lambda)$ for Layers A and B, as observed by Cooper et al. (2021). We also find that the $\theta_z$ dependence of $\overline{k_e}(x)$ is small in the range of $\theta_z$ from 50– 86°, and thus, we determine the $\theta_z$-independent

$\overline{k_e}(x)$ values by averaging it over this range of $\theta_z$. By plotting the $\theta_z$-independent values of $\overline{k_e}(x)$ as a function of $x$ for clear and cloudy skies (Fig. 3), the $\overline{k_e}(x)$ values rapidly decrease in the shallower ice layer and change to a gentle slope in the deeper ice layer when $x$ increases. This is because the spectral distribution of $F_b(\lambda)$ in bare ice changes from a wider distribution to a narrower distribution centred around the minimum $k_e(\lambda)$ wavelength, where the light absorption by bare ice is weakest as $x$ increases. We can then obtain the approximation equations by fitting the calculated values of $\overline{k_e}(x)$ with a power regression equation, as shown in Table S1. In this study, we use the terms "$\kappa_{clr}$" and "$\kappa_{cld}$" as the $\overline{k_e}(x)$ values for clear and cloudy skies, respectively. Finally, we determine the following approximation equations (Eqs. 26 and 27) of the broadband flux extinction coefficients as a function of ice thickness for clear ($\kappa_{clr}$; m$^{-1}$) and cloudy ($\kappa_{cld}$; m$^{-1}$) skies (Eq. 25 and Fig. 3).

$$\kappa_{clr} = 1.917 D_{t-1}^{-0.613}, \tag{26}$$

$$\kappa_{cld} = 1.620 D_{t-1}^{-0.519}. \tag{27}$$

where, $\kappa_{clr}$ and $\kappa_{cld}$ are calculated using the Layer A values shown in Table S1, as we focus on the vertical change in cryoconite holes below 0.20 m. The broadband flux extinction coefficient for the diffuse component ($\kappa_f$) is determined in a similar manner for the estimation of $R_{sd}$ and $R_{sf}$ as follows:

$$\kappa_f = \left(1 - r_{dif}\right)\kappa_{clr} + r_{dif}\kappa_{cld}. \tag{28}$$

As there are no available data for estimating the extinction coefficient for the direct component ($\kappa_d$), we parameterize it by introducing a coefficient ($r_d$) as follows:

$$\kappa_d = r_d \kappa_f. \tag{29}$$

The coefficient $r_d$ was assumed to be 1/1.66. The value 1.66 is used as an approximation value of the effective optical path length when the transmittance of the diffuse component of shortwave radiation is obtained from that of the direct component of shortwave radiation (Liou, 1980). Because we first estimate the diffuse component coefficient, $\kappa_d$ was obtained by dividing $\kappa_f$ by 1.66.

## 3. Observations

### 3.1. Site location

To evaluate the model, we investigated changes in CH depth at the Qaanaaq ice cap in Northwest Greenland (Fig. 4) during the 2012, 2014, and 2017 summer seasons. The ice cap covers an area of 286 km$^2$ and has an elevation range of 0–1,200 m a.s.l. (Sugiyama et al, 2014). We selected five study sites at different elevations (Sites 1 to 5) on the Qaanaaq Glacier, which is an outlet glacier of the ice cap that is easily accessible from Qaanaaq Village. Sites 1 to 5 are located in the middle of the glacier at elevations of 247, 441, 672, 772 and 944 m a.s.l., respectively. The equilibrium line elevation of the glacier ranged between 862 and 1,001 m a.s.l. in early August 2012, 2014, and 2017 (Tsutaki et al., 2017); hence, the ice surfaces at Sites 1 to 5 were exposed during the ablation period from July to August.

### 3.2. Observations of meteorological variables and surface reflectance

The meteorological data that were used as input in CryHo were collected with an automatic weather station (AWS), which was established at the SIGMA-B site (77.518° N, 69.0619° W; 944 m a.s.l.), i.e., at Site 5 in this study, on 19 July 2012 (Aoki et al., 2014; Nishimura et al., 2021). Hourly air temperature ($T_a$; K), relative humidity ($h_r$; dimensionless), downward shortwave radiation ($R_S$; W m$^{-2}$), downward longwave radiation ($R_L$; W m$^{-2}$), upward longwave radiation ($R_{Lui}$; W m$^{-2}$), wind speed ($U$; m s$^{-1}$), and air pressure ($P_a$; hPa) were collected for the model simulations described in Sect. 4. The seasonal changes of the measured air and surface temperatures and the calculated daily surface energy balance during the three studied summer seasons are shown in Figure 5. The air and surface temperatures support the assumption that all study sites below the SIGMA-B site (i.e., Site 5) were mostly in ablation conditions during the three studied summer seasons. The heat balance was similar across the study years.

To determine the ice surface albedo used as input in CryHo, we measured the spectral reflectance from the visible to near-infrared band (0.350–1.050 μm) on the ice surface using a spectrometer (MS-720, Eiko Seiki Co., Japan) at the study sites during the 2014 and 2017 summer seasons (Table 2). For details on the observation method, we refer to Takeuchi et al. (2015). Although the mean reflectance differs optically from broadband albedo (0.300–3.000 μm), the reflectance measured at Site 3 in 2014 agrees well with the broadband albedo at the same site measured in 2012 (0.4 versus approximately 0.4; Takeuchi et al., 2018). Therefore, we assume that the reflectance shown in Table 2 can be used as a proxy for broadband albedo at the study sites for numerical simulations.

### 3.3. Monitoring cryoconite hole depth

To collect in situ data that can be used for the evaluation of CryHo, the temporal changes in CH depth were observed using the monitoring device, which consisted of two time-lapse cameras as well as a plastic stick positioned in the centre of a CH, supported by metal angles buried to approximately 1.5 m depth in the ice body (Fig. 6a). The temporal changes in CH depth were derived from variations in the heights of the plastic stick and white metal angle (Figs. 6b–d). To measure the variations,

we marked both the stick and angle with clear and black colours at 5 cm intervals. The stick and angle were used to estimate the melt rates at the CH bottoms and ice surface surrounding the CHs, respectively. The variations were measured from hourly images with a resolution of 4,608 × 2,592 pixels captured by a time-lapse camera (PENTAX WG-10, RICOH Co., Japan) installed at 3 m from the marked stick and angle. The time-lapse camera was placed 1 m above the ice surface, while being orthogonally oriented to the ice surface. To maintain the camera at the same height from the ice surface, we adjusted the height regularly (i.e., every 5–10 days). To capture the CH collapse event and check whether the plastic stick remained positioned in the CH, we also monitored the CHs using hourly images with a resolution of 1280 × 1022 pixels taken by another time-lapse camera (GardenWatch Cam, Brinno Co., Taiwan, Fig. 6b). The location of the device was selected to have a suite of characteristics broadly identical to those of the monitoring site in a previous study (Takeuchi et al., 2018) in terms of cryoconite loading, ice structure, and ice surface slope of ~5°. Monitoring was conducted from day of the year (DOY) 194 to 214 in 2014.

To estimate the CH depths, the images were processed using ImageJ (version 1.48, National Institutes of Health, USA; Schneider et al., 2012). The CH depths were measured as relative variations using the length of marks on the stick for the CH bottom (Fig. 6c) and on the angle for the ice surface (Fig. 6d). One pixel corresponds to approximately 1 mm. Because the monitored CH collapsed between DOY 202 and 204, the stick and angles were reinstalled into another CH on DOY 205 (Fig. 6e).

Camera-based observations were conducted at Site 2, near the site where CH collapse occurred in 2012 (Takeuchi et al., 2018). We also conducted manual measurements of the depth and diameter of CHs two to four times during the 2014 and 2017 summer seasons. In addition, we used the dates of the CH collapse events reported by Takeuchi et al. (2018) for model evaluation. CH depths and diameters were manually measured from 20 to 50 randomly selected CHs at each site. Averages and standard deviations of the observed depths and diameters are summarised in Table 2.

## 4. Experimental design of the model evaluation and sensitivity tests

To evaluate CryHo, numerical simulations of the CH depth were conducted at several sites of the Qaanaaq ice cap for the three summer seasons (Site-exp). The number of study sites in 2012, 2014, and 2017 were one (Site 3), four (Sites 1 to 4), and three (Sites 1 to 3), respectively. The depths and diameters observed on the first observational date were used to set the model constants at each site ($D_0$ and $\phi$ in Table 2). The meteorological data measured at Site 5 were used as input for the simulations. At each site, $T_a$ was corrected based on the air temperature measured at Site 5 and a temperature lapse rate of $7.80 \times 10^{-3}$ K m$^{-1}$, which had been previously estimated from field observations on the same glacier in 2012 (Sugiyama et al., 2014). The other meteorological data at each site were assumed to be the same as those measured at Site 5. The observed surface temperature at Site 5 was almost the melting point during the three observational periods; thus, it was safely used also as input for the other sites, since they were located at lower elevations than that of Site 5 (Fig. 5a). In 2014, $\alpha_i$ at Sites 1, 2, 3, and 4 was assumed to be 0.68, 0.57, 0.40, and 0.41, respectively (Table 2), as mentioned in Sect. 3.2. Similarly, in 2017, $\alpha_i$ at Sites 1, 2, and 3 was assumed to be 0.56, 0.42, and 0.24, respectively (Table 2). $\alpha_i$ in 2012 for Site 3 was assumed to be 0.4, as observed by

Takeuchi et al. (2018). We assumed the albedo at the CH bottom ($\alpha_c$) to be 0.1 based on assumptions and observations in previous studies (e.g., Takeuchi et al., 2001; Jepsen et al., 2010). The calculation periods for the simulations in 2014 and 2017 were taken from the first date when CH depths were measured in June or July. In the 2012 simulation, the period was constrained by both CH and AWS observations.

We conducted sensitivity tests to assess the sensitivity of the CH depth to input data and model constants, such as air temperature ($T_a$-exp), radiation components ($R_S$-exp), initial depth ($D_0$-exp), hole diameter ($\phi$-exp), albedo at the ice surface ($\alpha_i$-exp), albedo at the CH bottom ($\alpha_c$-exp), extinction coefficients of direct ($\kappa_d$-exp) and diffuse ($\kappa_f$-exp) radiation, solar zenith angle ($\theta_z$-exp), and zenith angle of the edge from the centre of the CH bottom ($\theta_c$-exp) (Table 3). Site-exp, i.e., Site 2 in 2014, was used as the control experiment for the sensitivity tests (Ctl-exp). The ranges of the changing parameters, which are summarised in Table 3, were determined based on field measurements (Table 2). The extinction coefficients for $\kappa_d$-exp and $\kappa_f$-exp were obtained from multiplying by factors of 0.25–4.00 the original values. The factor range was assumed by referring to the difference between the spectral flux extinction coefficient and absorption coefficient calculated from the imaginary refractive index of pure ice (Fig. 2). In $R_S$-exp, we assumed $r_{dif}$ of Eqs (9) and (10) to be 0 and 1 in Sd and Sf cases shown in Table 3, respectively. Because we did not consider the light refraction at the air–water surface in the CH, we evaluated the refraction effect via sensitivity analysis ($\theta_z$-exp) using various incident angles ($\theta_z$). Although the heat balance at the CH bottom should vary across the width of the hole, particularly between the northern and southern edges, we assumed heat balance only at the centre of the CH bottom for the model simulation. The effect of different edges is evaluated by changing $\theta_c$ ($\theta_c$-exp). In $\theta_z$-exp and $\theta_c$-exp, $\theta_z$ and $\theta_c$ calculated in the model were replaced with the values shown in Table 3, respectively.

To quantify the four components of shortwave radiation reaching the CH bottom ($R_{Sdc}$, $R_{Sfc}$, $R_{Stdc}$, and $R_{Stfc}$) for different CH geometries (i.e., depths and diameters), we conducted two sensitivity tests ($R_{Sc}D$-exp and $R_{Sc}\phi$-exp, respectively) at 13:00 and 1:00 local time on DOY 172 (i.e., the time of meridian transit and midnight at the summer solstice) in 2014. The CH depth and diameter ranged from 0.01 to 0.35 m in $R_{Sc}D$-exp and from 0.01 to 0.15 m in $R_{Sc}\phi$-exp, respectively. The CH diameter and depth were assumed to be 0.05 m in $R_{Sc}D$-exp and 0.01 m in $R_{Sc}\phi$-exp, respectively. The other model constants were the same as those in Ctl-exp.

To discuss the meteorological factors controlling the CH dynamics, besides the sensitivity tests described above, we also investigated the relationships between the CH deepening or shallowing rates against the input meteorological conditions, and the simulated heat properties at the ice surface at the CH bottom. These relationships were obtained from the input and output data in Ctl-exp. The deepening (shallowing) rate (m h$^{-1}$) was calculated when a CH deepened (shallowed) compared to its state one hour earlier.

## 5. Results and discussion

### 5.1. Evaluation of the cryoconite hole depth

The CH depth simulations at the different elevation sites in the three years (Site-exp) showed that the simulated temporal changes in CH depth, including CH collapse, agreed well with those observed at all sites for the three years (Figs. 7 and 8). The CH depth observed by both the camera and manual measurements at Site 2 gradually shallowed from DOY 186 to 202, developed until DOY 212, and shallowed again until DOY 215 (Fig. 7b). The modelled and camera-based CH depths at this site were in good agreement with each other ($R^2 = 0.78$; root mean square error of 0.04 m). In addition, the model also simulated the CH collapse event between DOY 202 and 204, during which the camera-based CH depth was not recorded (line gap in Fig. 7b). The model simulations revealed the good performance of the model also at the other sites (Figs. 7a, c, and d). Although no CH depth measurement was conducted in 2012, the model captured the CH collapses observed at Site 3 on DOY 206 and 208 (Fig. 8a, grey dashed lines). The manual measurements in 2017 showed similar temporal changes in CH depth to those in 2014 (Figs. 8b–d), which were also well represented by the model, especially the timing of drastic CH decays around DOY 210 at all sites. Because CryHo diagnoses the CH depth based on the balance between melt at the ice surface ($M_i$) and at the CH bottom ($M_c$), the model can capture the CH decay dynamics, which have not been simulated by previous numerical models considering only $M_c$ (Gribbon, 1979; Jepsen et al., 2010; Cook, 2012). This indicates that the concept of CryHo, in which both heat balances (i.e., $Q_i$ and $Q_c$) control the temporal change in the development and decay of CHs, is essentially correct. In the next section, we discuss in detail the meteorological and/or ice physical factors that are important for simulating temporal changes.

### 5.2. Sensitivity of the simulated hole depth to the model parameters

The sensitivity experiment regarding the air temperature ($T_a$-exp) suggests that CHs tend to be deeper and more stable under cooler conditions, and vice versa (Fig. 9a). This is likely because the turbulent heat fluxes on the ice surface decrease under cooler conditions; thus, net radiation becomes the main component for ice melting, which is constrained by the albedo at the ice surface and CH bottom. Indeed, field observations indicated that the turbulent heat fluxes during warmer conditions (DOY 198–205) were increased compared to the case during cooler conditions (before DOY 198) at Site 2 in 2014 (Fig. 5b). Under such cooler conditions, the melt rate at the ice surface surrounding the CHs was lower, resulting in deeper and therefore more stable CHs. Indeed, Gribbon (1979) reported that the CH depth increased with the elevation of the glacier in West Greenland. Furthermore, higher melt rates (hence, warmer conditions) at the ice surface are likely to induce CH decay, as suggested by Hodson et al. (2007). Therefore, global warming may cause the collapse of CHs earlier in the melt season compared to the present case, thereby inducing darkening of the ice surface by cryoconite granule redistribution in the future. Nevertheless, a snow–ice–atmosphere coupling simulation using a climate model coupled with CryHo is necessary to evaluate the contribution of CH dynamics to surface darkening under climate change.

The sensitivity experiment regarding the shortwave radiation components ($R_S$-exp) suggests that CHs tend to decay and develop when the direct and diffuse components, respectively, are dominant (Fig. 9b). We then compared the contributions of each radiation component reaching the CH bottom during the experimental period (Fig. 10, red and blue lines). The CH bottom is more accessible by the diffuse component ($R_{Sfc} + R_{Stfc}$) rather than the direct component ($R_{Sdc} + R_{Stdc}$), except for shallowing depth case. In the model, the direct component of shortwave radiation can reach the CH bottom only when the solar

zenith angle $\theta_z$ is smaller than the CH edge $\theta_c$ (Fig. 1 and Eq. 15) and it is transmitted through the ice in the opposite case. Because $\theta_z$ and $\theta_c$ ranged from 56 to 85° and 8 to 90° during the simulation period at the studied glacier, respectively, the direct component reaching the CH bottom from the hole mouth was very limited. Figure 11a shows that there is no significant difference between the direct and diffuse components reaching the CH bottom even at the time of meridian transit in the summer solstice, suggesting that the diffuse component reaches the CH bottom more frequently than the direct component in

the studied glacier. To investigate heat flux to the CH bottom by shortwave radiation from the hole mouth or through the ice, we additionally compared the contributions of each radiation component reaching the CH bottom (Fig. 10, grey lines). The figure depicts that the radiation components transmitted through ice to the CH bottom ($R_{Stfc} + R_{Stdc}$) was greater than the radiation components reaching the CH bottom from the hole mouth ($R_{Sfc} + R_{Sdc}$) when CH developed from DOY 208 to 213, meaning that radiation components transmitted through ice are also important for the heat balance at the CH bottom (i.e., $Q_c$).

Further discussion regarding shortwave radiation transmitted throughout the ice is described later.

      The sensitivity experiments regarding the CH geometry ($D_0$-exp and $\phi$-exp) suggest that differences in the geometrical parameters have little influence on the CH depth. The simulated CH depth starting from different initial depths converged within approximately two weeks (Fig. 9c). This is probably because the CH bottom is relatively accessible by shortwave radiation in the case of shallower depths, and vice versa (Fig. 11), which is quantified by additional experiments ($R_{Sc}D$-exp

and $R_{Sc}\phi$-exp). The diameter shows no significant effect on the CH depth (Fig. 9d), although it strongly affects whether shortwave radiation reaches the CH bottom. This is likely because transmitted shortwave radiation reaches the CH bottom irrespective of the CH diameter. Figure 11b suggests no significant difference in the total shortwave radiation reaching the CH bottom among the different diameters, thereby supporting the $\phi$-exp result. Indeed, no significant correlation between the CH depth and diameter has been found (Gribbon, 1979; Cook, 2012). Because an increase in CH diameter allows more direct

shortwave radiation to reach the CH bottom, positive feedback of the CH development is possible. However, such feedback is unlikely to occur because Figures 11b and d suggest that most of the diffuse component of the transmitted shortwave radiation reaches the CH bottom at 0.01 m depth in CHs over 0.03 m in diameter. Furthermore, the observed CH depths and diameters significantly varied among the sites and years (Table 2), suggesting that CH depth is mainly controlled by factors other than the CH diameter.

The sensitivity experiments regarding the albedos of the ice surface and CH bottom ($\alpha_i$-exp and $\alpha_c$-exp) suggest that the difference between ice surface ($\alpha_i$) and CH bottom albedo ($\alpha_c$) is an important factor for reproducing the CH dynamics, especially the timing of CH collapse. The CH depth increases with an increase in $\alpha_i$ owing to decreasing $M_i$, whereas it

decreases with an increase in $\alpha_c$ owing to decreasing $M_c$ (Figs. 9e and f). Notably, the sensitivity of the CH depth to $\alpha_i$ was greater than that to $\alpha_c$. This is probably because shortwave radiation at the ice surface was greater than that at the CH bottom.

In addition, $\alpha_i$-exp suggests that a 0.1 decrease in $\alpha_i$ induces CH collapse one day earlier. Although $\alpha_c$ is known to be a key parameter for simulating the CH deepening rate (Podgorny and Grenfell, 1996), there is little information and discussions regarding $\alpha_i$. Because we used constant values of $\alpha_c$ and $\alpha_i$ in $\alpha_i$-exp and $\alpha_c$-exp, respectively, the CH depth sensitivity to the albedo is equivalent to the sensitivity difference between $\alpha_i$ and $\alpha_c$. Therefore, our results first highlight the importance of the difference between $\alpha_i$ and $\alpha_c$ in simulating vertical CH variations in which CH tends to develop with greater albedo difference and vice versa. Furthermore, the collapse of CHs is likely to reduce the ice surface albedo, owing to the redistribution of cryoconite on the ice surface (Irvine-Fynn et al., 2011; Takeuchi et al., 2018). Considering studies that reported a darkening trend of the GrIS surface over the last 20 years (e.g., Shimada et al., 2016; Tedstone et al., 2017), a positive albedo feedback, during which a decrease in ice surface albedo causes frequent CH collapse events, may occur. While our assumed reflectance was based on the $\alpha_i$ observations in 2014, the ice surface albedo should change temporally. The uncertainty of the model simulations, such as that regarding the results around DOY 204 at Site 3 in 2017 (Fig. 8d), may be attributed to temporal changes in the ice surface albedo. To simulate the development and decay of CHs more accurately, temporal changes in the ice surface albedo, including the effects of CH collapse and algal blooms, should be incorporated into CryHo.

The sensitivity experiments regarding the broadband extinction coefficients of shortwave radiation transmitted throughout ice for the direct and diffuse components ($\kappa_d$-exp and $\kappa_f$-exp) suggest that $\kappa_f$ is more effective at the CH depth than $\kappa_d$. Both experiments showed that the CH depth with a higher coefficient was shallower, owing to a decrease in $M_c$ (Figs. 9g and h); however, there was a significant difference in the sensitivity to the coefficients. One reason is probably that $\kappa_d$ is lower than $\kappa_f$, as in Eq. (29). Figure 10 showed that the diffuse component of shortwave radiation reaching the CH bottom was greater than the direct component of that even though $\kappa_f$ is higher than $\kappa_d$, suggesting that the shortwave radiation diffuse component reaches the CH bottom more easily than the direct component. Indeed, Figure 11 indicates that the reaching fraction of direct component of shortwave radiation transmitted throughout the ice to the CH bottom decreases with $\theta_z$. This is probably the cause of the higher sensitivity to $\kappa_f$, suggesting that $\kappa_f$ is likely a relatively more important parameter compared to $\kappa_d$. However, there was no significant difference in sensitivity between $\kappa_f$ and $\kappa_d$ in the case of $\theta_z \leq \theta_c$, which is the case for depths shallower than approximately 0.50 m in Figures 9g and h. In such a case, most of the direct and diffuse components reach the CH bottom without being transmitted through the ice (Fig. 11). Figure 11 also suggests that shortwave radiation transmitted throughout the ice is dominant in the opposite case. Since CH depth ranged from 0 to 0.20 m at Site 2 in 2014, the result supports our argument that transmitted shortwave radiation mainly contribute to CH development as described in the paragraph of $R_S$-exp. In Antarctic, a lot of CHs of over 0.30 m depth were reported even though the CH mouths were covered with frozen ice (Fountain et al., 2004), suggesting that the contribution of the transmitted shortwave radiation to CH development. The results highlight the importance of separating the direct and diffuse components of shortwave radiation in CH depth simulations. Further studies on the optical characteristics of ice are necessary, because there is little information on

the broadband flux extinction coefficients. Furthermore, previous studies reported the development of a porous ice layer known as the weathering crust on glaciers worldwide during summer (Irvine-Fynn and Edwards, 2014; Stevens et al., 2022). The weathering crust density is sometimes less than 500 kg m$^{-3}$ (Müller and Keeler, 1969). Because the shortwave radiation transmittance would be larger in a low-density case than that in the ice density case assumed in this study (900 kg m$^{-3}$), CHs may also develop under the weathering crust. To accurately simulate temporal change in CH depth, more detailed surface ice properties such as the weathering crust layer should be considered for future CH modelling.

The sensitivity experiments regarding the zenith angle ($\theta_z$-exp and $\theta_c$-exp) suggest that differences in the zenith angles have little influence on CH depth, except for the instance in which the downward shortwave radiation always reaches the CH bottom from the hole mouth ($\theta_z = 0°$). $\theta_z$-exp showed that CH depth with a higher $\theta_z$ was shallower, owing to a decrease in $M_c$ (Fig. 9i), and $\theta_c$-exp showed that CH depth with a lower $\theta_c$ was also shallow (Fig. 9j). Notably, the experiments suggest that both $\theta_z$ and $\theta_c$ hardly affect CH depth at over 15° and below 60°, respectively. Snell's law states that the direct component of incident radiation is refracted through the air–water surface. According to this law, the refraction angle is approximately 20° smaller than the incident angle $\theta_z$, therefore the direct component of the downward shortwave radiation more easily reaches the CH bottom from the hole mouth. However, such refraction is unlikely to affect CH depth because $\theta_z$ is always greater than 55° at the studied latitude. Although CryHo calculates $M_c$ at the centre of the CH bottom using $\theta_c$, $M_c$ may differ at the northern and southern edges of the CH bottom because the zenith angles of the CH bottom edges would differ from $\theta_c$. $\theta_c$-exp suggests that CH depth is temporally non-uniform on the northern and southern edges of the CH bottom. However, CH depth likely becomes uniform again over time according to $D_0$-exp. Indeed, the simulated CH depth using a different $\theta_c$ converged within approximately two weeks (Fig. 9j). A previous study suggests that the surface of the CH bottom on steep north-sloping ice is non-uniform due to heterogeneous radiation reaching the CH bottom (Cook et al., 2018). Because the slope and aspect in our studied sites are below 5° and southwest, respectively, the surface of the CH bottom might remain uniform on such gentle south-sloping ice.

### 5.3. Meteorological factors controlling the cryoconite hole dynamics

The sensitivity tests in the previous section showed that CH depth did not equilibrate in any experimental cases at the studied glacier, where meteorological conditions change before the depth reaches equilibrium (Fig 9). It suggests that meteorological conditions significantly affect CH depth. For example, downward shortwave radiation governs the CH deepening processes (Hodson et al., 2010a; Takeuchi et al., 2018). To better understand the factors controlling the CH deepening and shallowing processes, we compared the heat components with both the CH deepening and shallowing rates. In the shallowing phase (red marks in Fig. 12), CH decay is mainly controlled by the sensible and latent heat fluxes at the ice surface ($H_{Si}$ and $H_{Li}$) via wind speed ($U$). Air temperature ($T_a$) shows no significant correlation with the shallowing rate; conversely, $U$ shows a strong correlation with the shallowing rate (Figs. 12a and d). However, there is no clear relationship between the input downward shortwave radiation, input downward longwave radiation, relative humidity, and direct and diffuse components of downward

shortwave radiation at the ice surface ($R_S$, $R_L$, $h_r$, $R_{Sd}$ and $R_{Sf}$ shown in Figs. 12b, c, e, g, and h, respectively). The results suggest that strong winds induce CH decay by increasing the surface melt via the turbulent heat fluxes (Figs. 12f–j). Takeuchi et al. (2018) suggested that CHs tend to be shallower under both cloudy and windy conditions. Our analyses also suggest that windy conditions are important meteorological conditions governing the CH decay (Fig. 13a). Because the total downward shortwave is reduced through clouds, although the diffuse component of downward shortwave radiation generally increases under cloudy conditions, turbulent fluxes are likely to be dominant in the total energy budget, resulting in an ice surface that melts faster than the CH bottom. This is probably the reason why the CH collapse events were observed under cloudy conditions.

In the CH deepening phase (blue marks in Fig. 12), although the input shortwave radiation ($R_S$) was weakly correlated with the CH deepening rate, the diffuse component reaching the CH bottom was strongly correlated with the CH deepening rate ($R_{Sfc} + R_{Stfc}$), whereas no correlation was found with the direct component reaching the CH bottom ($R_{Sdc} + R_{Stdc}$) (Figs. 12l–m). Furthermore, Figure 12k, which shows the relationship between ice melt at the CH bottom ($Q_c$) and the CH deepening rate, is similar to Figure 12m. The results suggest an essential contribution of the diffuse component to CH development, which was also revealed by sensitivity analysis. However, the direct component strongly depends on the solar zenith angle and CH geometry ($\theta_z$ and $\theta_c$); thus, the CH development is likely to be unstable during sunny conditions. Other meteorological variables are unlikely to contribute to the CH development. Although previous studies suggested that sunny conditions induce CH development (Hodson et al., 2010b; Irvine-Fynn et al., 2011; Takeuchi et al., 2018), our results highlight that the diffuse component of shortwave radiation is a key factor governing CH development, rather than merely sunny conditions. Because the CH deepening rate is likely to depend on wind speed and the intensity of the diffuse component of downward shortwave radiation, as shown in Figure 13b, CHs could develop under sunny conditions depending on wind speed and other meteorological conditions. CH depths are mainly determined by the balance between the intensity of the diffuse component of downward shortwave radiation and the strength of the turbulent heat transfer.

## 5.4. Other possible factors affecting cryoconite hole dynamics

Our model does not include the effect of water lingering in CHs on the heat balance at the CH bottom because a quantitative understanding of the mechanism of convective heat transport or the buffering effect in the lingering water is insufficient. Such lingering water in CHs may affect the heat exchange between the atmosphere and CH bottom. Heat exchange should not be negligible in the case of large water surfaces in CHs. Although the water level in CHs is not estimated in the model, the refraction through the air–water surface in CHs is unlikely to affect CH depth in the studied glacier as discussed regarding $\theta_z$-exp. However, such refraction might contribute to CH development in lower latitude regions such as Asia, where the solar zenith angle is significantly smaller than that in the polar region. In addition to the refraction, reflectance at the water surface would reduce the amount of shortwave radiation reaching the CH bottom. To simulate CH depths globally, such an effect may need to be better incorporated into CryHo. Besides lingering water in CHs, the thickness of cryoconite at the CH bottom, which

485 is not considered by CryHo, is also likely to be a key factor in determining the CH diameter and shape because a portion of the absorbed radiation in the cryoconite at the CH bottom could be transferred laterally and then melt the CH wall (Cook et al., 2010; Cook, 2012). Furthermore, the surface slope of the glacier affects the sediment thickness and, thus, the CH shape geometry via water flow and changes the shortwave radiation reaching the CHs (Takeuchi et al., 2000; Cook et al., 2018). Although such factors should be incorporated into CryHo in the future, CryHo is likely to reproduce well the temporal change 490 in CH depth, even for relatively large CH on flat surfaces with diameters up to around 0.1 m, which falls within the range of CH shape geometries documented in this study. Compared to the effects of CH shape geometry and topological conditions, the effects of lingering water and sediment thickness in CHs on the CH depth may be small. The CH diameter range of 0.01 to 0.11 m assumed for $\phi$-exp is similar to that observed in glaciers in West Greenland, Svalbard, and the Himalayas (Gribbon, 1979; Takeuchi et al., 2000; Cook et al., 2010), suggesting that CryHo can be applied also to these regions.

## 495 6. Conclusions

We established a numerical cryoconite hole model (CryHo) to reproduce the vertical cryoconite hole (CH) dynamics. We evaluated the model using field observations from an ice cap in Northwest Greenland, collected during the 2012, 2014, and 2017 summer seasons. The CryHo simulates the temporal changes in CH depth based on heat balances calculated independently at the ice surface and the CH bottom. This is a novel concept that is different from previous models simulating 500 the CH development. CryHo considers also the CH geometry, which affects the direct and diffuse components of shortwave radiation, while including the optical process of transmitted radiation throughout ice. Field observations revealed that CHs repeatedly developed and decayed over time. CryHo reasonably reproduced the observed temporal changes in CH depth, including the CH shallowing phase at four sites during three melt seasons. Sensitivity tests indicated that the CH bottom is more accessible by the diffuse component— rather than the direct component—of shortwave radiation, thereby controlling the 505 CH development. In addition, both components of shortwave radiation transmitted throughout ice also significantly contribute to CH development, except for shallowing CH or large diameter cases. Besides wind speed, which has been indicated by earlier studies, we revealed that the difference in albedo between the ice surface and the CH bottom is also an important factor affecting the timing of the CH collapse. Although many studies in Greenland have reported the bio-albedo effect contributing to ice surface darkening, CH collapse could also reduce the ice surface albedo not only by redistributing the cryoconite granules, 510 but also by increasing the phototrophic (i.e., algae, cyanobacteria, diatoms, etc.) and heterotrophic (i.e., tardigrade, rotifer, etc.) communities. Therefore, CHs dynamics are interactively connected to physical and microbiological processes on the ice surface. For improved projection of spatiotemporal changes in the bio-albedo effect, CryHo will be useful when coupled with regional or global climate models, although additional observations and model evaluation in various glaciers and ice sheets are necessary.

## Code and Data availability

The codes of the model as well as the analysis and observational data used in this study are available at the following reference: https://doi.org/10.5281/zenodo.7968848 (Onuma et al., 2023). Meteorological observations from the SIGMA-B site are available at https://ads.nipr.ac.jp/data/meta/A20220413-006/ (Nishimura et al., 2021).

## Author contributions

KF designed the study. YO, NT, and TA designed the field observations. KF developed the model with the support of MN and TA. TA parameterized the extinction coefficients of ice. All authors performed field observations. YO, KF and TA analysed the data and wrote the manuscript. All authors contributed to the discussion.

**Competing interests.** Masashi Niwano is a member of the editorial board of the journal The Cryosphere.

## Acknowledgements

This work was supported by the Arctic Challenge for Sustainability (ArCS, JPMXD130000000), Arctic Challenge for Sustainability II (ArCS II, JPMXD1420318865), and Advanced Studies of Climate Change Projection (SENTAN, JPMXD0722680395) from the Ministry of Education, Culture, Sports, Science and Technology (MEXT), Japan. This study was also partly supported by Grant-in-Aids (JP23221004, JP26247078, JP26241020, JP16H01772, JP16H06291, JP19H01143, JP20K19955, JP21H03582 and JP23K17036). We thank J. Uetake, N. Nagatsuka, R. Shimada, S. Tanaka, R. Sakaki, K. Ishiwatari, and A. Watanabe for their support with the field observations.

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

**Table 1: Variables and constants used in this study. Variables having different surfaces are indicated by an asterisk in the subscript, $i$ indicates ice surface and $c$ indicates cryoconite hole (CH) bottom. $^+$ denotes the input data for the cryoconite hole model (CryHo).**

| Symbol | Variable or constant | Unit and value |
|--------|---------------------|----------------|
| $\alpha_i$ | Surface albedo at ice surface | 0.24 – 0.68 (see Table 2) |
| $\alpha_c$ | Surface albedo at CH bottom | 0.1 |
| $C$ | Bulk coefficient for snow-ice surface | 0.0025 |
| $c_p$ | Specific heat of air | 1006 J K$^{-1}$ kg$^{-1}$ |
| $D_0$ | Initial depth | m |
| $D_t$ | Depth of cryoconite hole | m |
| $F_b(\lambda)$ | Spectral attenuated solar radiation | W m$^{-2}$ |
| $F_s(\lambda)$ | Spectral solar radiation | W m$^{-2}$ |
| $H_{L*}$ | Latent heat flux | W m$^{-2}$ |
| $H_{S*}$ | Sensible heat flux | W m$^{-2}$ |
| $h_r$ | Relative humidity$^+$ | - |
| $k_a(\lambda)$ | Bare ice spectral flux absorption coefficient | m$^{-1}$ |
| $k_e(\lambda)$ | Bare ice spectral flux extinction coefficient | m$^{-1}$ |
| $\overline{k_e}(x)$ | Broadband flux extinction coefficient | m$^{-1}$ |
| $\kappa_d$ | Broadband flux extinction coefficient of ice for direct component | m$^{-1}$ |

| | | |
|---|---|---|
| $\kappa_f$ | Broadband flux extinction coefficient of ice for diffuse component | $m^{-1}$ |
| $\kappa_{clr}$ | Broadband flux extinction coefficient of ice for clear sky | $m^{-1}$ |
| $\kappa_{cld}$ | Broadband flux extinction coefficient of ice for cloudy sky | $m^{-1}$ |
| $\lambda$ | Wavelength | $\mu m$ |
| $l_E$ | Latent heat for water evaporation | $2.50 \times 10^6 \text{ J kg}^{-1}$ |
| $l_M$ | Latent heat for ice melting | $3.33 \times 10^5 \text{ J kg}^{-1}$ |
| $M_*$ | Melting ice thickness | $m \text{ h}^{-1}$ |
| $P_a$ | Air pressure[+] | hPa |
| $\rho_a$ | Air density | $kg \text{ m}^{-3}$ |
| $\rho_i$ | Ice density | $900 \text{ kg m}^{-3}$ |
| $Q_{M*}$ | Excess heat energy for ice melting | $W \text{ m}^{-2}$ |
| $Q_*$ | Heat balance | $W \text{ m}^{-2}$ |
| $q(T)$ | Saturated specific humidity | $kg \text{ kg}^{-1}$ |
| $R_S$ | Downward shortwave radiation[+] | $W \text{ m}^{-2}$ |
| $R_L$ | Downward longwave radiation[+] | $W \text{ m}^{-2}$ |
| $R_{Lui}$ | Upward longwave radiation[+] | $W \text{ m}^{-2}$ |
| $R_{Ln*}$ | Net longwave radiation | $W \text{ m}^{-2}$ |
| $R_{Lc}$ | Longwave radiation from the sky looking up from the CH bottom | $W \text{ m}^{-2}$ |

| | | |
|---|---|---|
| $R_{Lw}$ | Longwave radiation from the CH wall | W m$^{-2}$ |
| $R_{Sd}$ | Direct component of shortwave radiation | W m$^{-2}$ |
| $R_{Sf}$ | Diffuse component of shortwave radiation | W m$^{-2}$ |
| $R_{Sdc}$ | Direct component of shortwave radiation reaching the CH bottom | W m$^{-2}$ |
| $R_{Sfc}$ | Diffuse component of shortwave radiation reaching the CH bottom | W m$^{-2}$ |
| $R_{Stdc}$ | Direct component of shortwave radiation transmitted through ice to the CH bottom | W m$^{-2}$ |
| $R_{Stfc}$ | Diffuse component of shortwave radiation transmitted through ice to the CH bottom | W m$^{-2}$ |
| $r_d$ | $\kappa_f$ to $\kappa_d$ conversion coefficient | 1/1.66 |
| $r_{dif}$ | Diffuse ratio of downward shortwave radiation | - |
| $r_{ze}$ | Ratio based on solar zenith angle | - |
| $r_{cld}$ | Ratio based on cloudiness | - |
| $T_a$ | Air temperature[+] | K |
| $T_i$ | Ice surface temperature | K |
| $T_c$ | CH wall or bottom temperatures | K |
| $T_s$ | Surface temperature derived from the automatic weather station | K |
| $T(\lambda, x)$ | Spectral flux transmittance as a function of ice thickness | - |
| $\bar{T}(x)$ | Spectrally integrated flux transmittance | - |
| $t_h$ | Length of an hour in seconds | 3600 s |

| | | |
|---|---|---|
| $U$ | Wind speed[+] | m s$^{-1}$ |
| $x$ | Ice thickness | m |
| $\varepsilon$ | Emissivity of the ice surface | 1.0 |
| $\theta_z$ | Solar zenith angle | radian |
| $\theta_c$ | Zenith angle of the edge from the centre of the CH bottom | radian |
| $\sigma$ | Stefan–Boltzmann constant | $5.67 \times 10^{-8}$ W m$^{-2}$ K$^{-4}$ |
| $\phi$ | Hole diameter | m |


**Table 2: Descriptions of cryoconite holes (CH) and ice surface reflectance at the study sites in 2012, 2014, and 2017. CH depths and diameters were observed using a ruler.**

| Site (year) | Elevation (m a.s.l.) | Day of year | Number of samples | Depth (mean $\pm$ SD m) | Diameter (mean $\pm$ SD m) | Ice surface reflectance (Observational date) |
|---|---|---|---|---|---|---|
| 3 (2012) | 672 m | 186 | 30 | 0.04 $\pm$ 0.01 | 0.03 $\pm$0.01 | No data |
| 1 (2014) | 247 m | 186 | 31 | 0.14 $\pm$ 0.02 | 0.04 $\pm$ 0.02 | 0.68 $\pm$ 0.03 (DOY 176) |
| 2 (2014) | 441 m | 186 | 31 | 0.19 $\pm$ 0.03 | 0.05 $\pm$ 0.02 | 0.57 $\pm$ 0.03 (DOY 176) |
| 3 (2014) | 672 m | 205 | 50 | 0.09 $\pm$ 0.02 | 0.03 $\pm$ 0.02 | 0.40 $\pm$ 0.07 (DOY 203) |
| 4 (2014) | 772 m | 205 | 50 | 0.11 $\pm$ 0.02 | 0.04 $\pm$ 0.02 | 0.41 $\pm$ 0.08 (DOY 203) |
| 1 (2017) | 247 m | 189 | 20 | 0.17 $\pm$ 0.03 | 0.05 $\pm$ 0.02 | 0.56 $\pm$ 0.11 (DOY 211) |
| 2 (2017) | 441 m | 189 | 20 | 0.15 $\pm$ 0.03 | 0.04 $\pm$ 0.02 | 0.42 $\pm$ 0.04 (DOY 211) |
| 3 (2017) | 672 m | 197 | 20 | 0.11 $\pm$ 0.03 | 0.04 $\pm$ 0.02 | 0.24 $\pm$ 0.06 (DOY 211) |


**Table 3: Overview of the sensitivity tests. Changing ranges for each variable are assumed based on the field observations in this study. Units of $\theta_z$ and $\theta_c$ in this table are converted to degrees for clarity.**

| | $\Delta T_a$ (K) | $R_S$ component | $D_0$ (m) | $\phi$ (m) | $\alpha_i$ (-) | $\alpha_c$ (-) | $\kappa_d$ ratio (times) | $\kappa_f$ ratio (times) | $\theta_z$ (degree) | $\theta_c$ (degree) |
|---|---|---|---|---|---|---|---|---|---|---|
| Values for control experiment | 3.92 | Both Sd and Sf | 0.19 | 0.05 | 0.57 | 0.1 | 1 | 1 | - | - |
| Experiment ID | $T_a$-exp | $R_S$-exp | $D_0$-exp | $\phi$-exp | $\alpha_i$-exp | $\alpha_c$-exp | $\kappa_d$-exp | $\kappa_f$-exp | $\theta_z$-exp | $\theta_c$-exp |
| Changing range | -3–+3* | Sd or Sf[+] | 0–0.2 | 0.01–0.11 | 0.3–0.7 | 0.05–0.3 | 0.25–4.0 | 0.25–4.0 | 0–75 | 15–90 |


\* change from the control experiment

[+] shortwave radiation assumed to be only direct or diffuse component

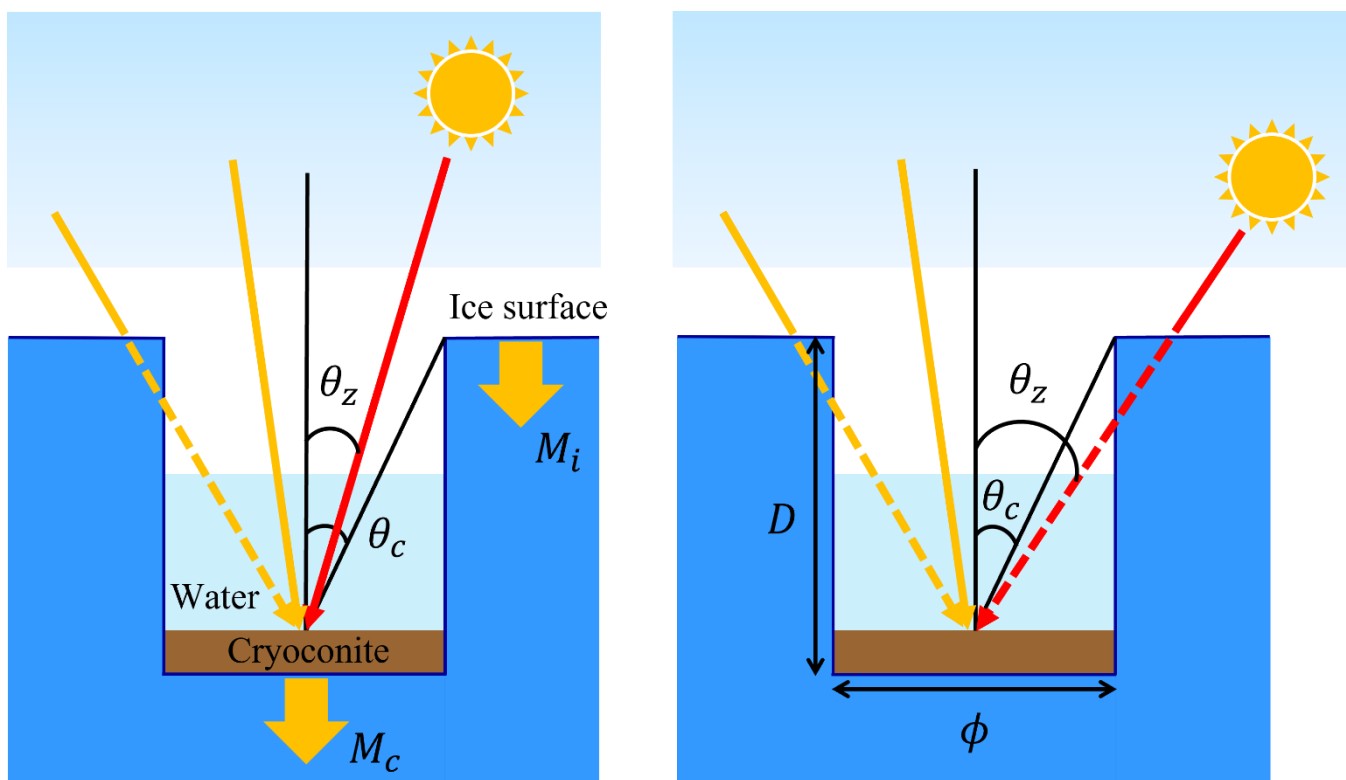

**Figure 1: Concept of the cryoconite hole model (CryHo). Heat balances at the surface and cryoconite hole bottom are independently calculated (left). Red and orange arrows indicate direct and diffuse components of shortwave radiation, respectively. Cryoconite hole (CH) geometry, with depth ($D$) and diameter ($\phi$) being considered for distinguishing the direct component of shortwave radiation (right). Cryoconite thickness at the CH bottom is assumed to be zero in the model. The difference between melting ice thickness at the surface ($M_i$) and that at the CH bottom ($M_c$) changes the CH depth. The direct component of solar radiation can**
**reach the CH bottom from the hole mouth if the solar zenith angle $\theta_z$ is smaller than the zenith angle of the CH edge $\theta_c$ (left, red solid arrow), while it is transmitted through the ice if the solar zenith angle is greater than the zenith angle of the CH edge (right, red dashed arrow). The diffuse component of downward shortwave radiation can reach the CH bottom regardless of $\theta_z$ (orange solid and dashed arrows).**

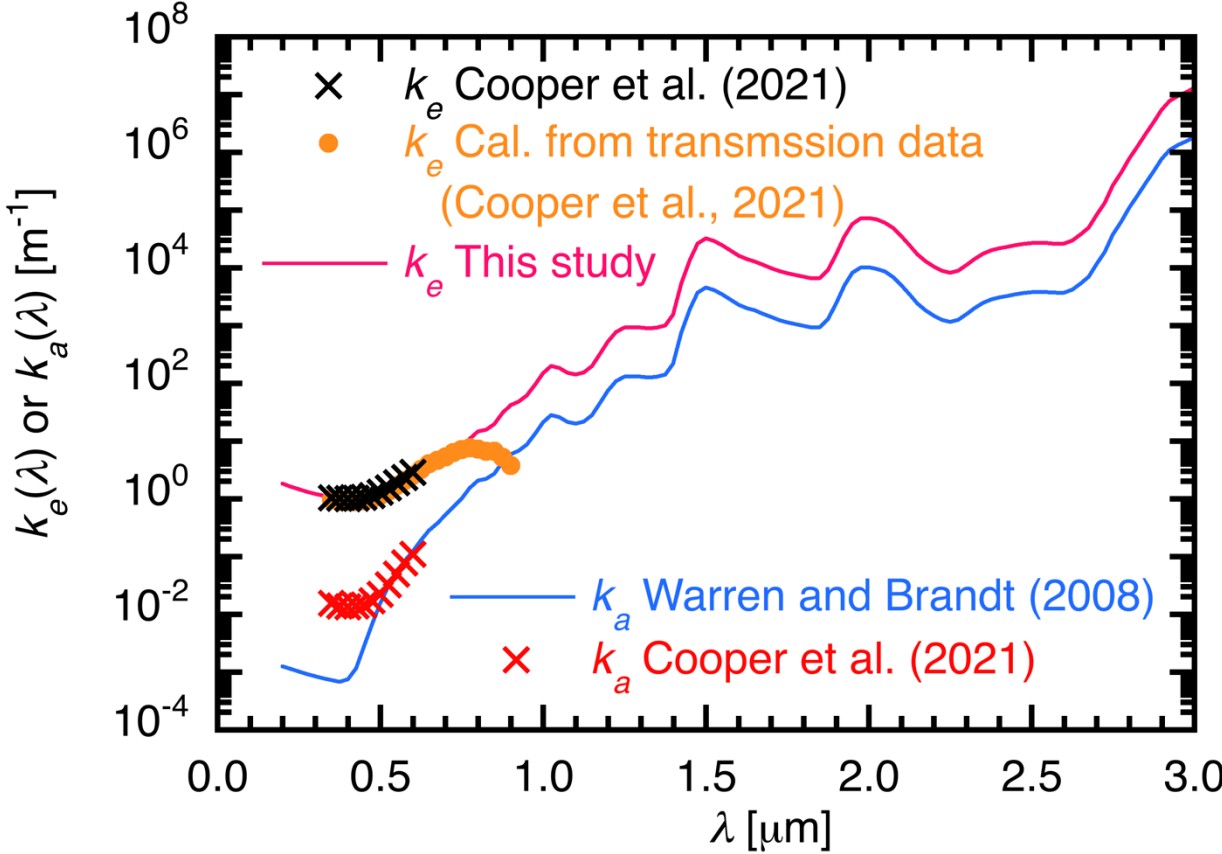

**Figure 2: Spectral flux extinction coefficient ($k_e(\lambda)$) and absorption coefficient ($k_a(\lambda)$) in $\lambda = 0.2$–$3.0$ μm used for the calculation of $\overline{k_e}(x)$. $k_e(\lambda)$ data for the wider spectral region in $\lambda = 0.2$–$4.0$ μm are used in the calculation of $\overline{k_e}(x)$. Black "x" symbols represent $k_e(\lambda)$ reported by Cooper et al. (2021) for Layer A, orange dots represent $k_e(\lambda)$ calculated from the transmittance data observed by Cooper et al. (2021), the pink curve is $k_e(\lambda)$ used in this study, the blue curve is $k_a(\lambda)$) calculated from the imaginary refractive index of pure ice (Warren and Brandt, 2008), and red "x" symbols represent $k_a(\lambda)$ shown in Figure S1 of Cooper et al. (2021) for Layer A, which is a reference for comparison with $k_a(\lambda)$ of Warren and Brandt (2008).**

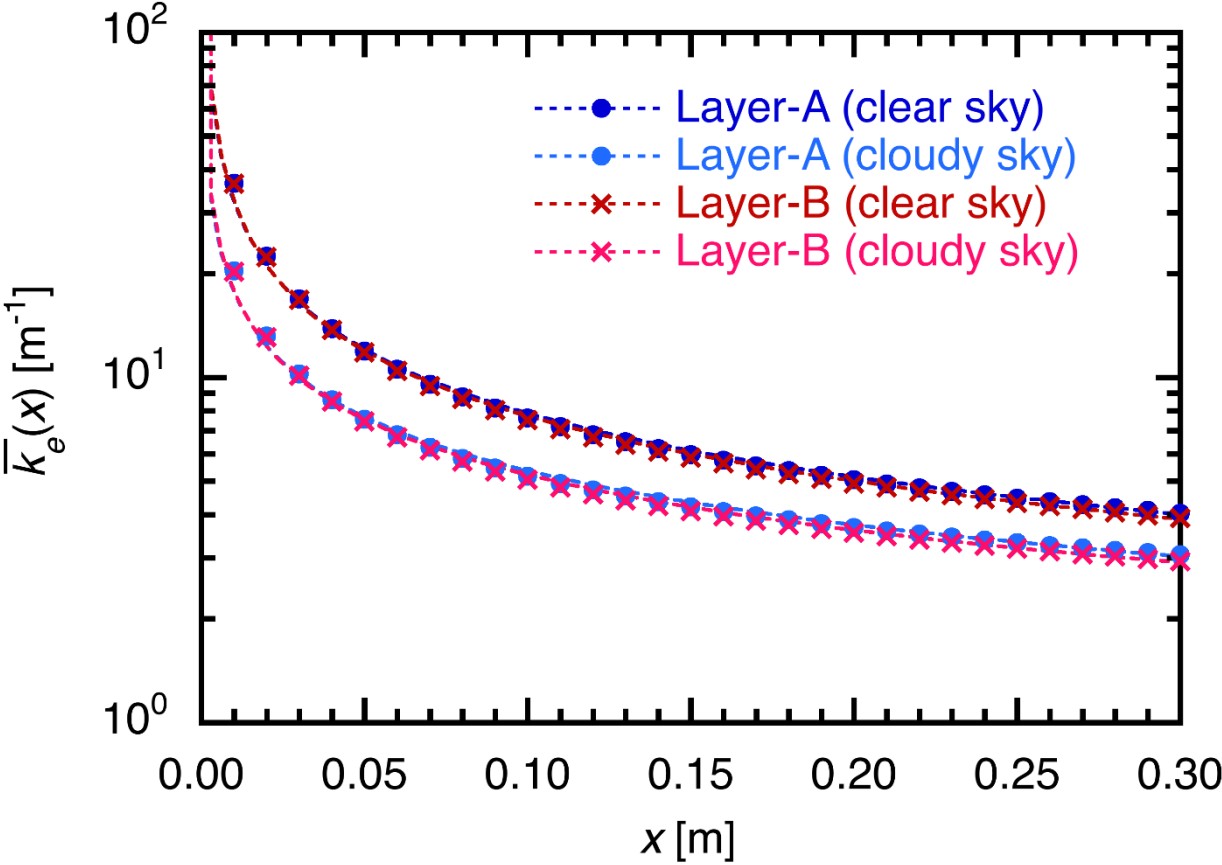

Figure 3: Broadband flux extinction coefficient $\overline{k_e}(x)$ of bare ice as a function of ice thickness $x$ for clear and cloudy sky conditions and employed data sets of $k_e(\lambda)$ for Layers A and B reported by Cooper et al. (2021). Calculated values are indicated by symbols and dashed curves represent approximation equations fitted by power regression equations.


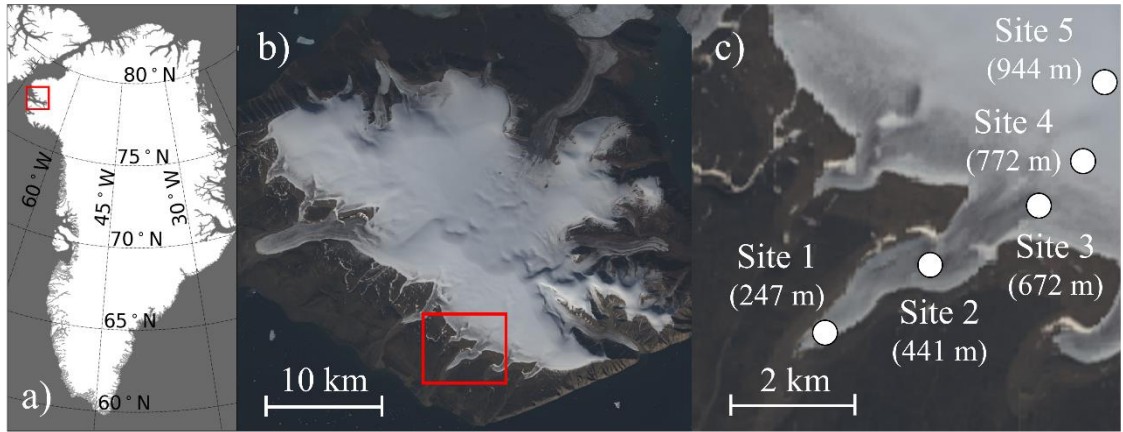


**Figure 4: (a) Map of Greenland, (b) Qaanaaq ice cap in Northwest Greenland, and (c) study sites along the Qaanaaq Glacier. Panels b and c are true-colour composite images of Sentinel-2 taken on 24 July 2017. Red rectangles in panels a and b indicate the Qaanaaq area and Qaanaaq Glacier, respectively.**

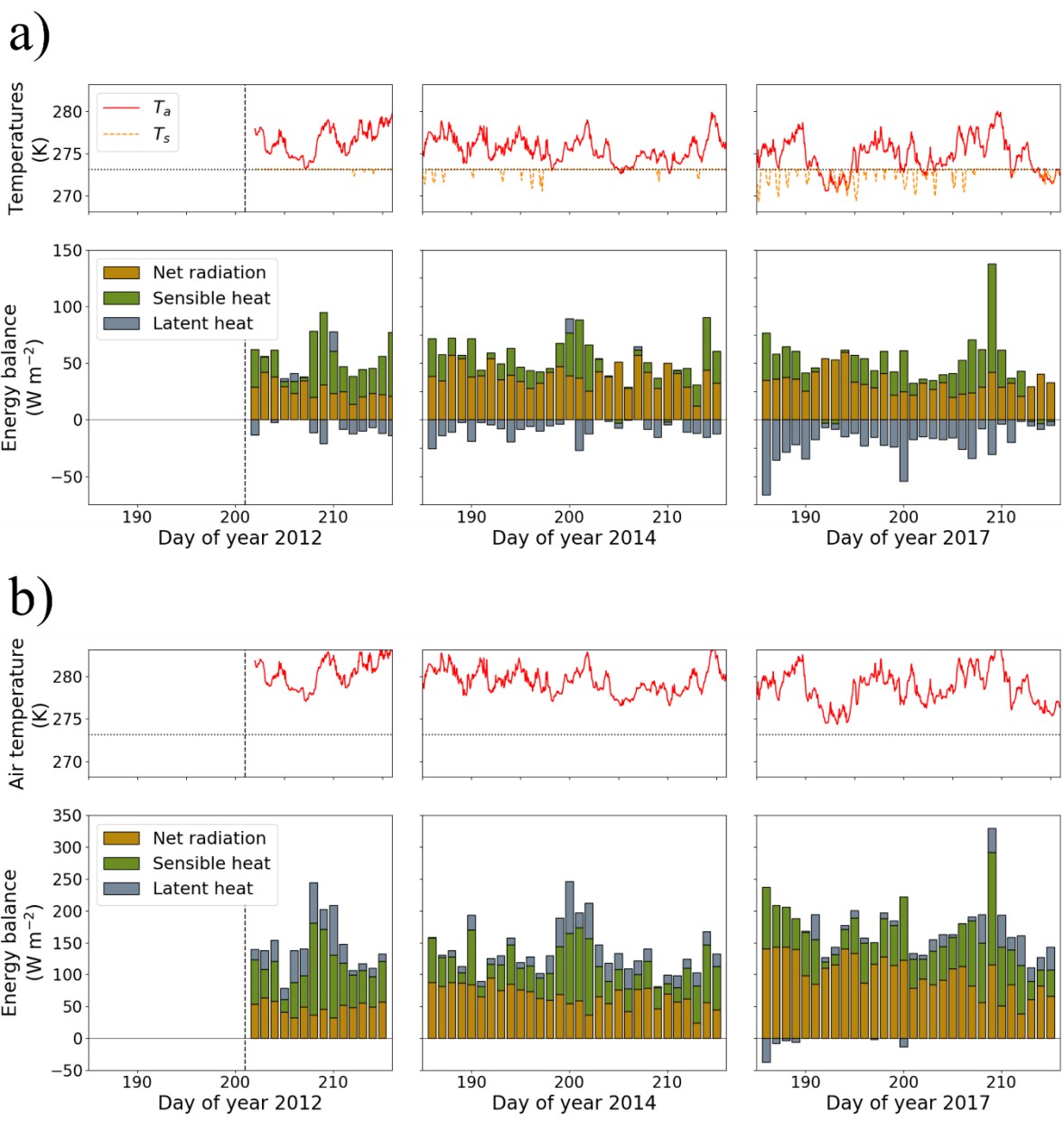

**Figure 5: (a)** Hourly air and surface temperatures ($T_a$ and $T_s$ in the upper panels, respectively) and daily surface energy balance (lower panels) at the Sigma-B site (Site 5 in this study) on the Qaanaaq ice cap during the 2012, 2014, and 2017 summer seasons from left to right, respectively. **(b)** Hourly air temperature (upper panels) and daily surface energy balance (lower panels) at Site 2. $T_s$ shown in the upper panels in (a) is calculated from the downward and upward longwave radiations. $T_a$ shown in the upper panels in (b) is corrected from $T_a$ in (a) and an observed lapse rate. Dotted lines in the upper panels indicate the melting point (273.15 K). Daily surface energy balance is calculated with CryHo. The individual energy flux positive signs show a downward component.

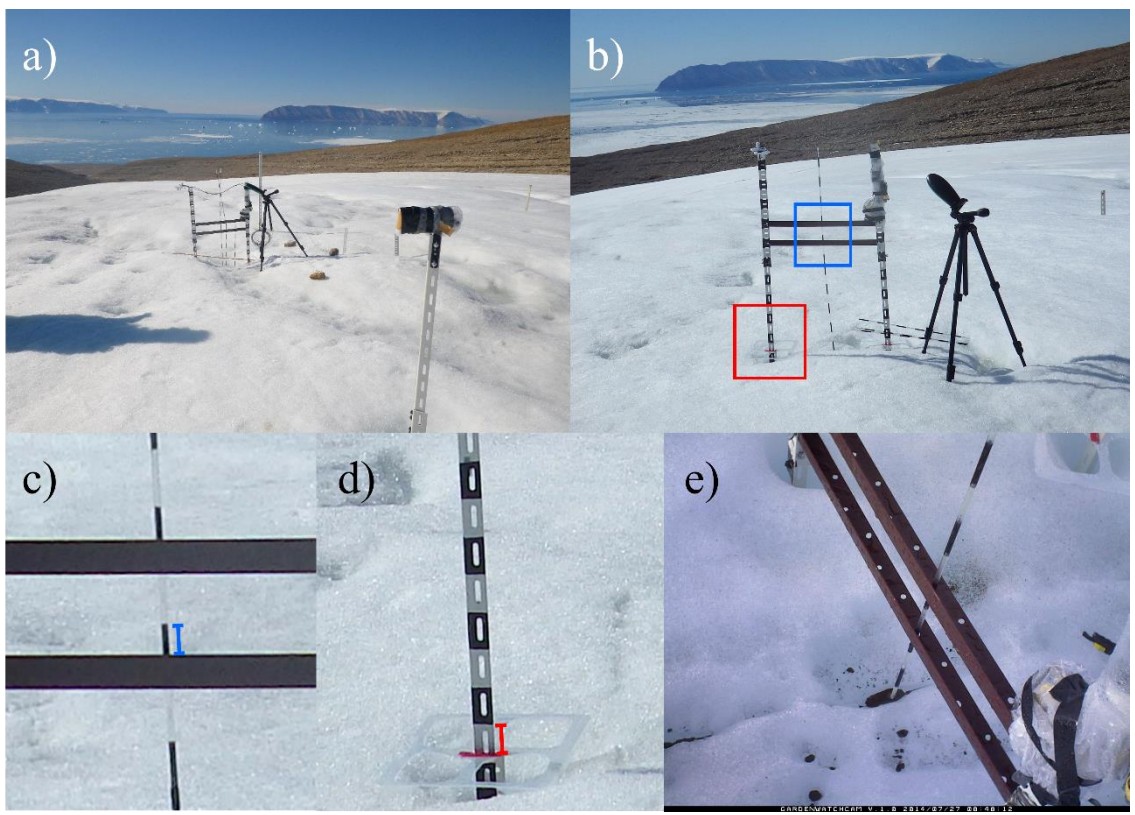

Figure 6: (a) Device for monitoring the cryoconite hole (CH) depth at Site 2, (b) sample image to quantify the CH depth, which was taken by the interval camera (PENTAX WG-10) on day of the year (DOY) 186 in 2014, (c) close-up of the image within the blue rectangle in panel b, (d) close-up of the image within the red rectangle in panel b, and (e) CH conditions monitored by the interval camera (GardenWatch Cam) on DOY 208 in 2014. Blue and red bars in panels c and d indicate measured lengths to estimate the vertical changes in CH bottom and ice surface, respectively. Because the ice surface was uneven, the ice surface level was inferred based on the red tape pasted on the clear plastic frame shown in panel d.

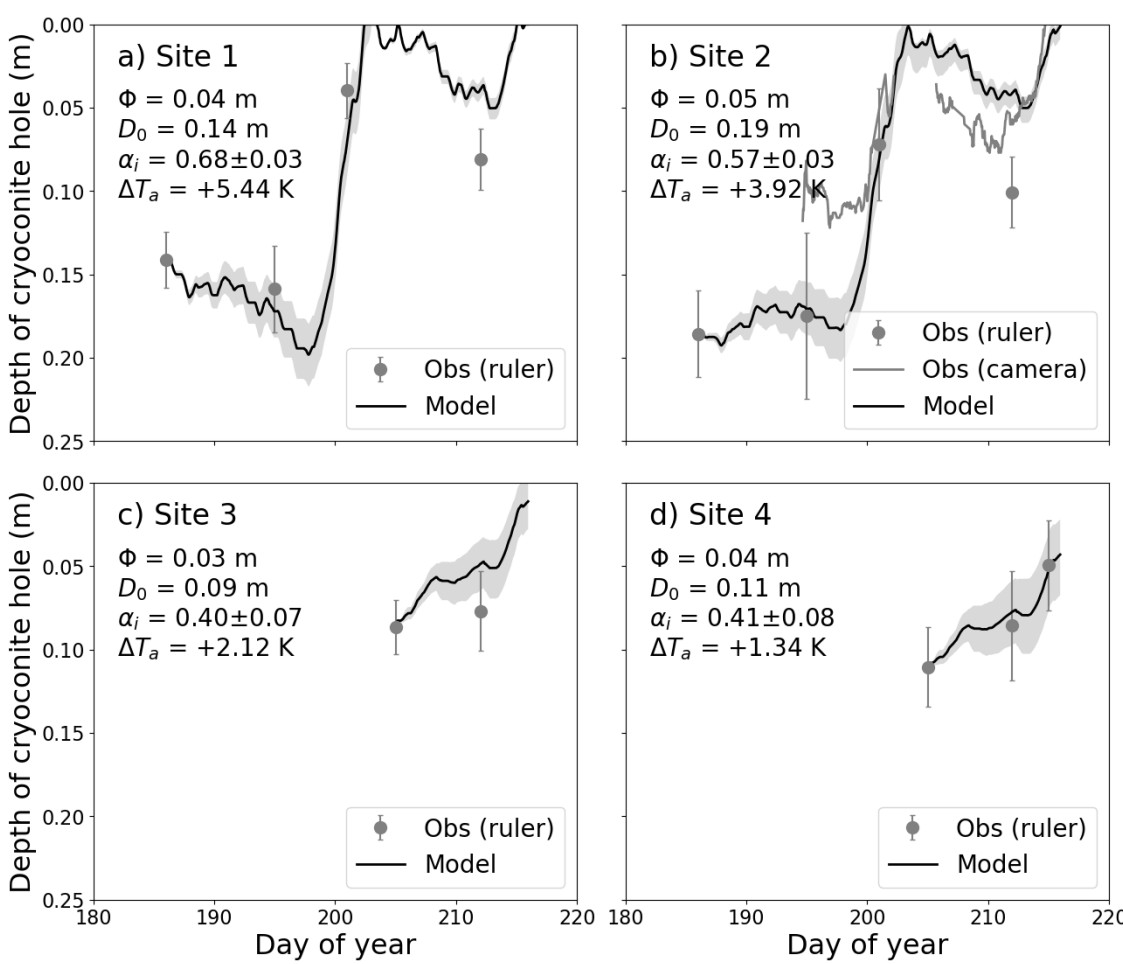

**Figure 7: Temporal changes in cryoconite hole (CH) depth at (a) Site 1, (b) Site 2, (c) Site 3, and (d) Site 4 in 2014. The CH constants, ice surface albedo and air temperature correction are described in each panel. The black solid line and shading indicate model results using average values and standard deviations of ice surface reflectance, which is assumed to be $\alpha_i$, as shown in Table 2. The missing grey solid line in panel b denotes the CH collapse period.**

750

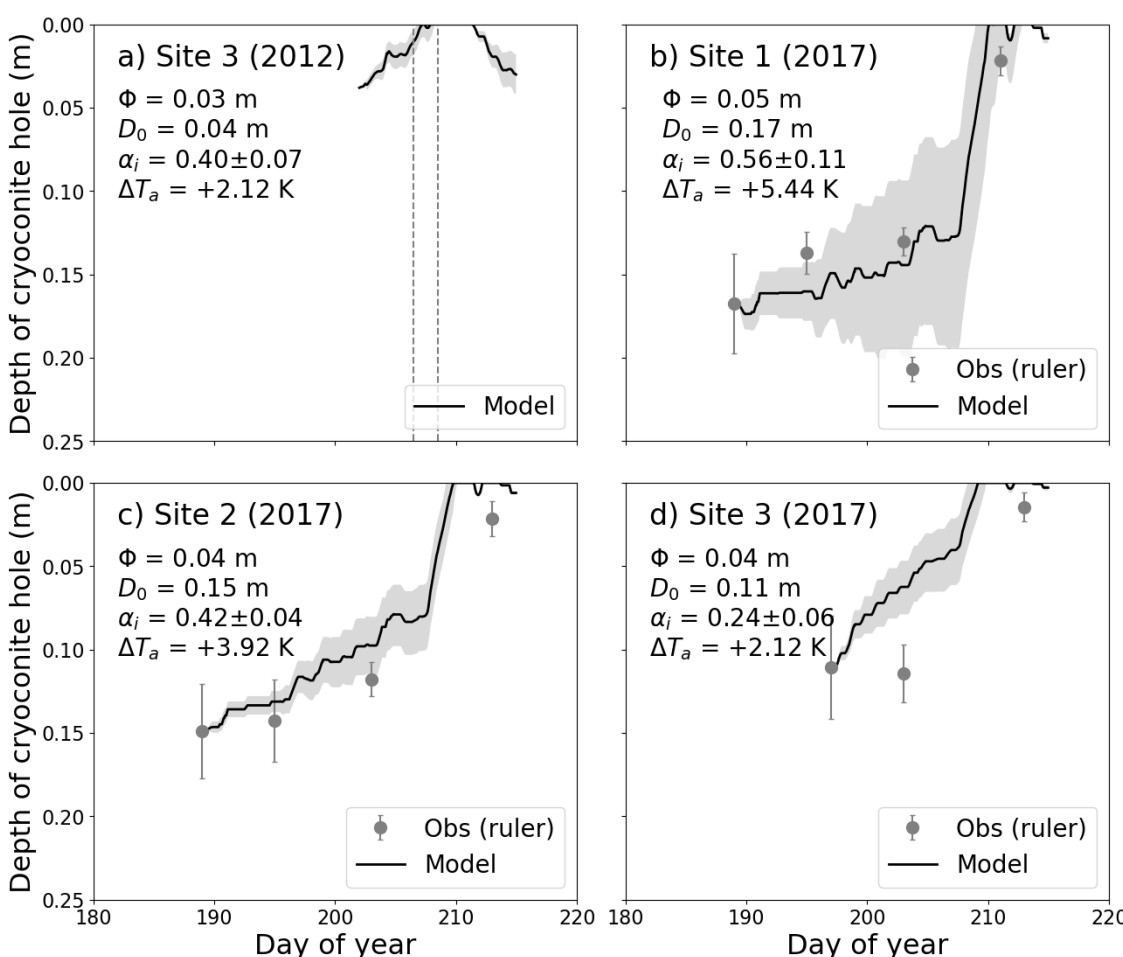

**Figure 8: Temporal changes in cryoconite hole (CH) depth at (a) Site 3 in 2012, (b) Site 1 in 2017 (c) Site 2 in 2017, and (d) Site 3 in 2017. The CH constants and air temperature correction are described in each panel. The black solid line and shading indicate model results using average values and standard deviations of ice surface reflectance, which is assumed to be $\alpha_i$, as shown in Table 2. Grey dashed lines in the panel a indicate the collapse dates reported by Takeuchi et al. (2018).**

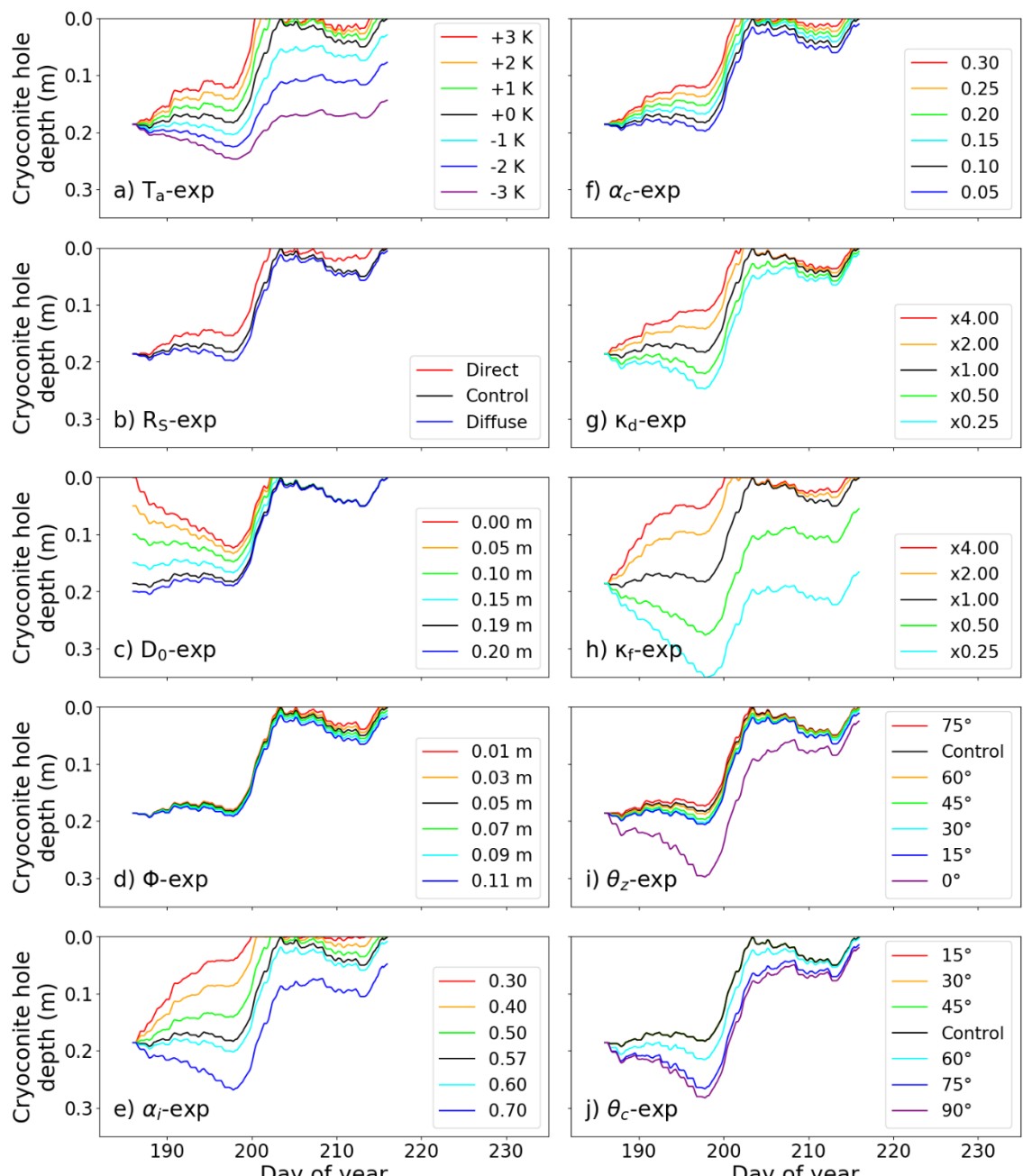

**Figure 9: Sensitivity experiments of the temporal changes in cryoconite hole (CH) depth to model parameters and meteorological conditions at Site 2 in 2014. (a) Air temperature ($T_a$-exp), (b) shortwave radiation ($R_S$-exp), (c) initial CH depth ($D_0$-exp), (d) CH diameter ($\phi$-exp), (e) ice surface albedo ($\alpha_i$-exp), (f) cryoconite albedo ($\alpha_c$-exp), broadband flux extinction coefficient of ice for the (g) direct component ($\kappa_d$-exp) and (h) diffuse component ($\kappa_f$-exp), (i) solar zenith angle ($\theta_z$-exp), and (j) zenith angle of the edge from the centre of the CH bottom ($\theta_c$-exp). Black lines in each figure indicate the control experiment ($Ctl$-exp). Note that lines for 15, 30 and 45° in the bottom right panel in (j) are overlapped with the line for Control.**

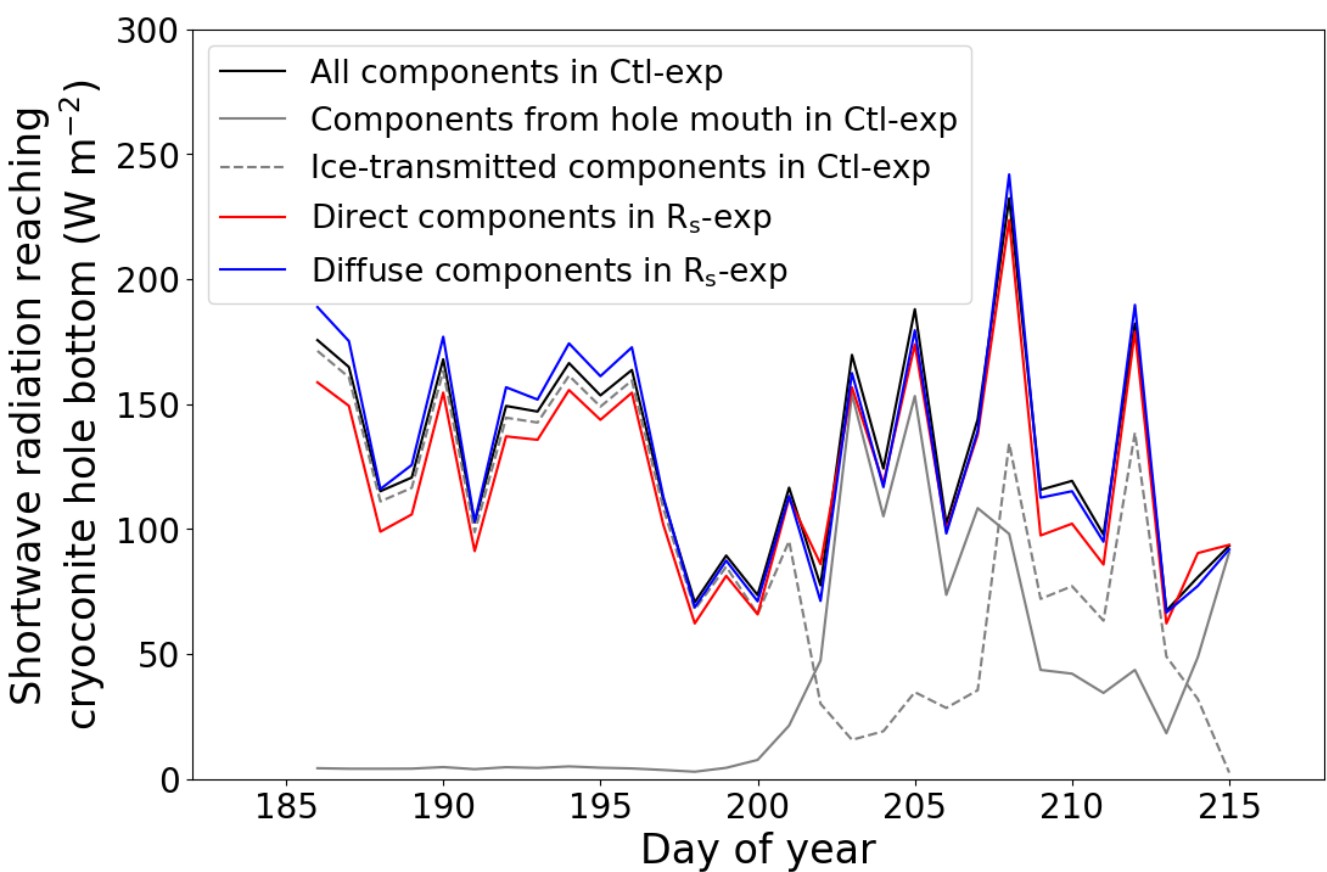

**Figure 10: Daily mean temporal changes in direct and diffuse components of shortwave radiation reaching the cryoconite hole (CH) bottom in 2014. Blue and red lines indicate the direct ($R_{sdt}+R_{stdc}$) and diffuse ($R_{sfct}+R_{stfc}$) components in $R_S$-exp, respectively. Black line indicates both component of shortwave radiation ($R_{sdt}+R_{sfct}+R_{stdc}+R_{stfc}$) in Ctl-exp. Grey solid and dashed lines indicate the radiation components reaching the CH bottom from the hole mouth ($R_{sdt}+R_{stdc}$) and transmitting through ice ($R_{sfct}+R_{stfc}$) in Ctl-exp, respectively.**

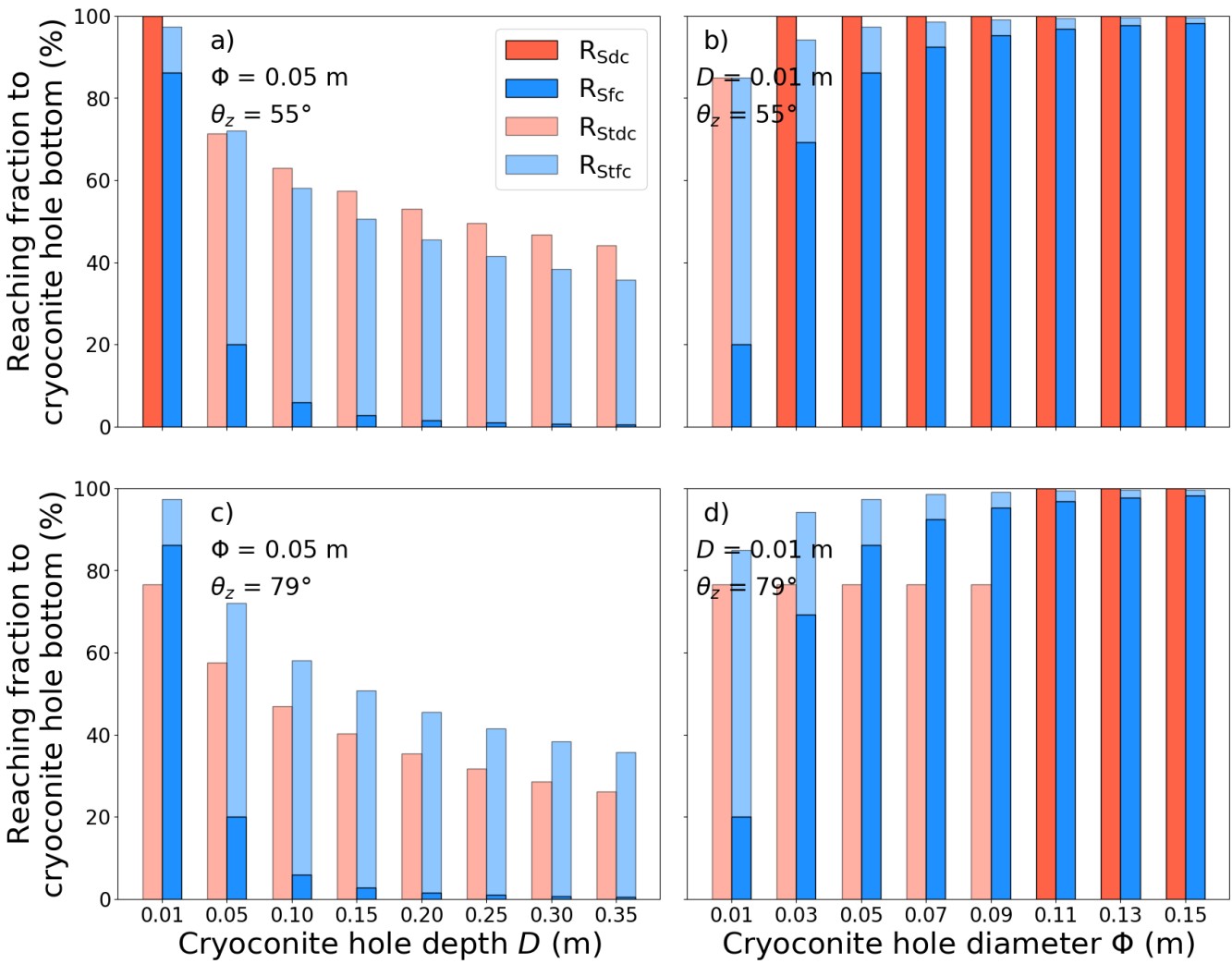

**Figure 11: Sensitivity experiments of the cryoconite hole (CH) geometry on the direct and diffuse components of shortwave radiation reaching the CH bottom for (a, c) CH depth and (b, d) CH diameter. Red and blue bars in the figure indicate the direct and diffuse components of shortwave radiation, respectively. Light red and blue bars indicate the direct and diffuse components of shortwave radiation transmitted throughout ice, respectively. The vertical axis represents the fraction of direct and diffuse components against the incoming shortwave radiation at the ice surface (100 % at the ice surface). The ratios were derived from numerical simulations with different CH depth or CH diameter on day of the year (DOY) 172 in 2014. The meteorological conditions for the simulations were assumed to be those at 13:00 local time (a, b) and 1:00 local time (c, d) on the date used for Ctl-exp.**

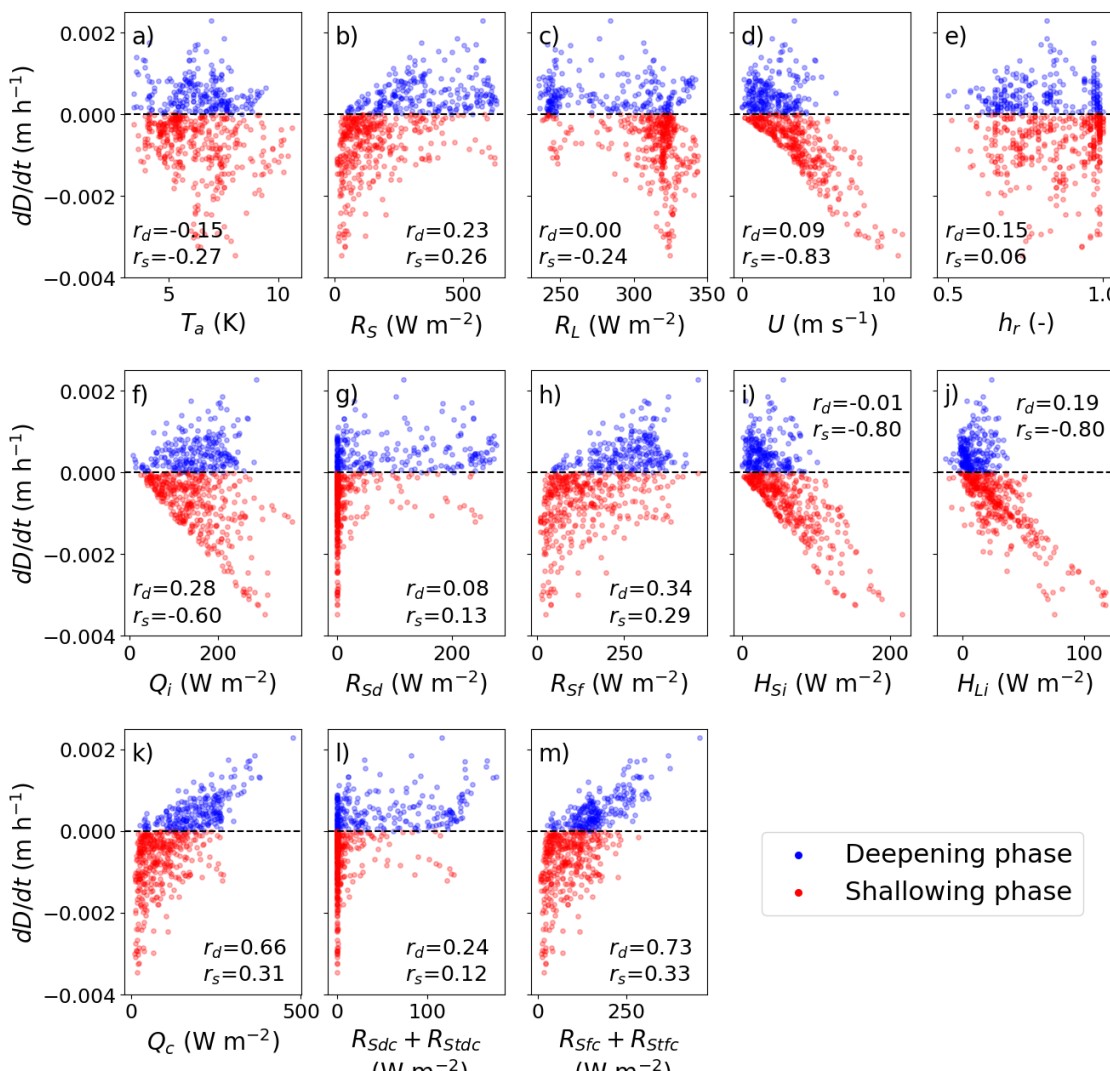

**Figure 12: Relationships between the CH deepening or shallowing rates in the control experiment (dD/dt, m h⁻¹). The upper (a–e), middle (f–j) and bottom (k–m) panels indicate the relationships between the CH deepening or shallowing rates and the input meteorological conditions, simulated heat properties at the ice surface, and simulated heat properties at the CH bottom, respectively. Blue and red marks indicate the CH deepening and shallowing, respectively. Variables in the horizontal axis are shown in Table 1. $r_d$ and $r_s$ in each panel indicate the correlation coefficients with CH deepening and shallowing rates, respectively. Note that $T_a$ in the panel a is air temperature corrected for Site 2 where the control experiment was performed.**

790

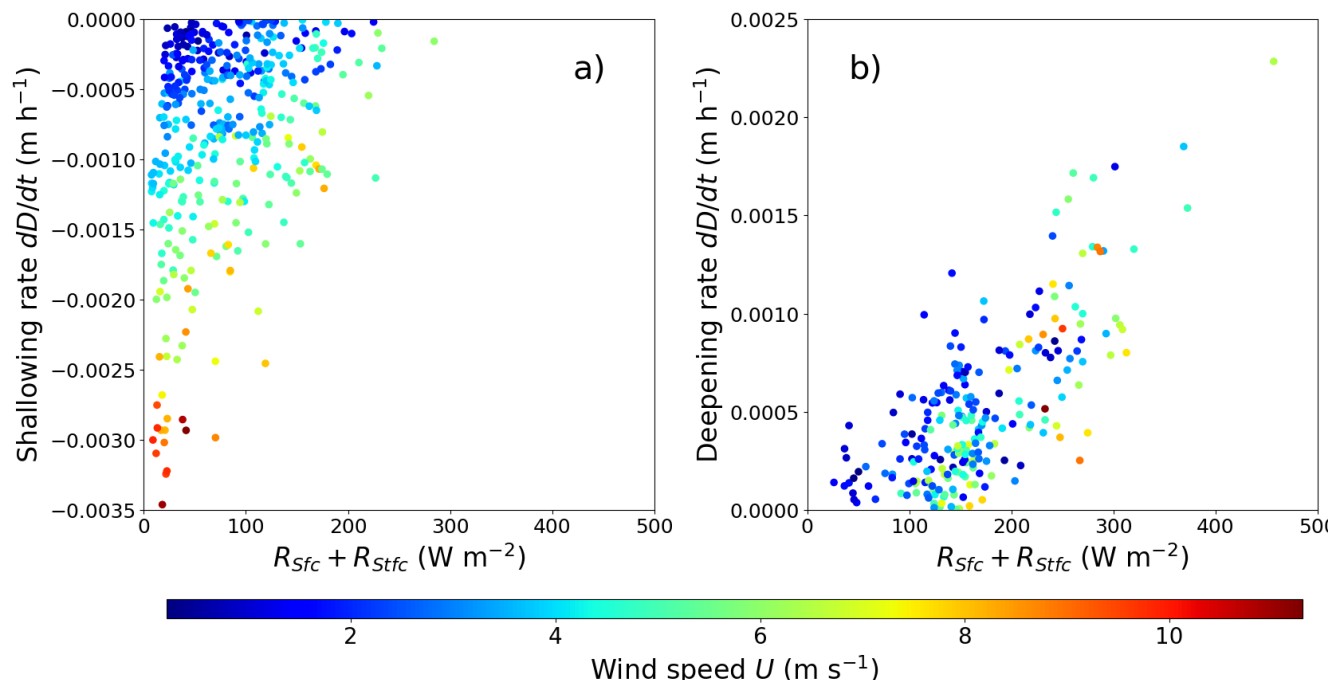

**Figure 13: Relationships between the cryoconite hole (CH) (a) deepening or (b) shallowing rates and the diffuse component of downward shortwave radiation reaching the CH bottom and wind speed in the control experiment. The CH shallowing and deepening phases shown in Figure 9m are separated into panels a and b in this figure, respectively. The contour in the panels indicates the input wind speed for the control experiment.**