# Peer review of "Modelling the development and decay of cryoconite holes in Northwest Greenland"

_EGUsphere, 2023_

## Author Comment (AC1)

**Reply to Dr. David Chandler (Referee#1)**

May 3, 2023
Dr. Yukihiko Onuma
Japan Aerospace Exploration Agency (JAXA)
E-mail: onuma.yukihiko@jaxa.jp

Dear Dr. Chandler,

   We would appreciate a number of valuable comments very much. Please see enclosed our responses to all your comments as well as the revised marked-up manuscript entitled "Modelling the development and decay of cryoconite holes in Northwest Greenland" by Yukihiko Onuma et al. [Paper #egusphere-2023-54] submitted to the journal The Cryosphere. Our responses (**blue text**) to each of your comments (**black text**) were described on the following pages. We also described the revised sentences with the yellow marker following your suggestions.

Best regards,
Yukihiko Onuma and co-authors

**RC1: 'Comment on egusphere-2023-54 (Yukihiko Onuma et al.)', David Chandler, 16 Mar 2023**

I'd like to thank the authors for their efforts developing this new model for cryoconite hole depth, which is well presented along with useful sensitivity experiments and some validation. As the authors point out, changes in cryoconite hole dynamics can influence ice surface albedo – so this is an important topic, given that SMB is one of the key controls on Greenland's sea-level contribution. I imagine this model could easily be driven by either AWS data or climate model output, making it a useful tool for investigating how cryoconite holes could influence albedo under climate warming anywhere in Greenland or indeed Antarctica given some basic observations of typical hole dimensions (which are already available for many places). Other applications would include supraglacial hydrology (changes in water storage) and ice surface microbial processes.

The model calculates changes in hole depth by considering energy balance at the centre of the hole. Validation with some field observations yields an encouraging match overall, with some discrepancies as we would expect.

There are two important aspects which I think need to be considered further before publication, given the application of this model in regions with generally large zenith angles. On that basis I have ticked the major revisions box, but I'm hoping it's not a lot of work to implement these changes. Elsewhere there are some minor points requiring additional clarification, and the manuscript needs language editing by a native English speaker as there are numerous grammatical errors and a few sentences which are a little hard to follow. Apart from the language itself, the paper is clear and easy to follow.

I'm not very up to date with the relevant literature so I just reviewed this study on its own merits and not in relation to other recent work.

We would like to thank you very much for taking the time to review our manuscript. Since the cryoconite hole model (CryHo) could be driven by climate model output as you mentioned, the model has a potential to spatio-temporally evaluate albedo reduction caused by collapses of CHs in the Greenland Ice Sheet under climate change. According to the suggestions from you and another reviewer, we have modified model code slightly, re-conducted numerical simulations including the sensitivity experiments, and revised the manuscript. The detailed our responses to your comments are as below.

**Main points**
(1) Refraction is not considered when the direct SW component passes from air to water. I wonder if

that would change your conclusion that the diffuse component dominates over the direct component. If the zenith angle (in the air) is theta_a, and the refractive indices of air and water are n_a = 1 and n_w = 1.33, then the zenith angle in the water (theta_w) would be estimated from Snell's law, i.e.,

n_a * sin(theta_a) = n_w * sin(theta_w).

This is worth considering, since your range of zenith angles in air (noted as 56 to 85deg: Line 305) would become 38 to 48deg in water, so it's much more likely that the direct SW can reach the hole centre. You might also want to consider reflection by the water surface.

Refraction along the transmitted (air-ice-water) pathway would also be worth considering but I imagine would be harder to implement.

I think it would be quite easy to adjust the model to account at least for this air-water refraction and hopefully not a lot of work to re-run the plotting scripts so we can see if this is important or not.

Thank you for your constructive comments. As you pointed out, the air-water refraction of light might increase the amount of solar radiation reaching the CH bottom even when the solar zenith angle is larger, while the opposite effect would occur when the reflection of light at the water surface reduces the amount of that reaching the CH bottom. Since these two effects depend on water depth, it is difficult to incorporate the effect of the refraction and reflectance into the model, which does not simulate water level in CH. However, we additionally conducted a sensitivity test to the solar zenith angle ($\theta_z$-exp) to discuss the effect of the zenith angle on the CH depth. The experiment showed that the solar zenith angle hardly affects CH depth in cases over 15° (Figure 9). This is probably due to that the direct component of downward shortwave radiation hardly reaches the CH bottom from the hole mouth in such cases. In the studied glacier, the solar zenith angle generally ranges from 56 to 85°, suggesting that the contribution of light refraction through air-water boundary to CH depth is insufficient. We add the result and discussion in Section 5.2 as well as the explanation about the sensitivity test in Section 4. The refraction may have better been considered to simulate CH depth globally because the solar zenith angle is sometimes below 20° in low latitudes such as Asia. This point has been raised in Section 5.4 as future challenge.

Regarding our conclusion that CHs tend to decay and develop in the case of that the direct and diffuse components are dominant, we have discussed the sensitivity of the CH depth to the shortwave radiation components in the paragraph for $R_S$-exp. Figures 9b and 10 in the revised manuscript showed that the CH tends to develop when the diffuse component is dominant. In the studied glacier, there is no significant difference between the direct and diffuse components reaching the CH bottom even at the time of meridian transit in the summer solstice (Figures 11a and 11c in the revised

manuscript). The result suggests that the diffuse component relatively reaches CH bottom more than the direct component in the studied glacier. This may be one of the reasons why the diffuse component contributes to CH development rather than the direct component in $R_S$-exp. We have added the result of $R_{Sc}D$-exp and $R_{Sc}\phi$-exp in the higher $\theta_z$ case (Figure 11) and the discussion in Section 5.2.

Lines 302-318:

[revised manuscript text omitted]

Lines 427-439:

The sensitivity experiments regarding the zenith angle ($\theta_z$-exp and $\theta_c$-exp) suggest that differences in the zenith angles have little influence on the CH depth, except for the case of that the downward shortwave radiation always reaches the CH bottom from the hole mouth ($\theta_z = 0°$). $\theta_z$-exp showed that the CH depth with a higher $\theta_z$ was shallower, owing to a decrease in $M_c$ (Fig. 9i). In contrast, $\theta_c$-exp showed that the CH depth with a lower $\theta_c$ was smaller (Fig. 9j). Notably, the experiments suggest that $\theta_z$ and $\theta_c$ hardly affect the CH depth in the case of over 15° and below 60°, respectively. Snell's law states that direct component of incident radiation is refracted through the air-water surface. The refraction angle is smaller approximately 20° than the incident angle $\theta_z$ by the law, therefore the direct component of the downward shortwave radiation more easily reaches the CH bottom from the hole mouth. However, such refraction is unlikely to affect the CH depth. Although the CryHo calculates $M_c$ at centre of the CH bottom using $\theta_c$, $M_c$ may differ at the northern and southern edges of the CH bottom because the zenith angles of the edges of the CH bottom would differ from $\theta_c$. $\theta_c$-exp suggests that the CH depth is a temporary non-uniform in the northern and southern edges of the CH bottom. However, the CH depth is likely uniform again over time according to $D_0$-exp. Indeed, the simulated CH depth using the different $\theta_c$ converged within approximately two weeks (Fig. 9j). In addition, CHs observed in the studied glacier were flat on the CH bottom.

Lines 471-479:

Our model does not include the effect of water lingering in CHs on the heat balance at the CH bottom because a quantitative understanding of the mechanism of convective heat transport or the buffering effect in the lingering water is insufficient. Such lingering water in CHs may affect the heat exchange

between the atmosphere and CH bottom. Heat exchange should not be negligible in the case of large water surfaces in CHs. Although water level in CHs is not estimated in the model, the refraction through air-water surface in CHs unlikely to affect CH depth in the studied glacier as discussed in $\theta_z$-exp. However, the refraction might contribute to CH development in the lower latitude regions such as Asia, where the solar zenith angle is significantly smaller than that in polar region. In addition to the refraction, reflectance at the water surface would reduce amount of shortwave radiation reaching the CH bottom. To simulate CH depths globally, such effects may have better been incorporated into CryHo.

[Figure]

Figure 9: Sensitivity experiments of the temporal changes in cryoconite hole (CH) depth to model parameters and meteorological conditions at Site 2 in 2014. (a) Air temperature ($T_a$-exp), (b) shortwave radiation ($R_S$-exp), (c) initial CH depth ($D_0$-exp), (d) CH diameter ($\phi$-exp), (e) ice surface albedo ($\alpha_i$-exp), (f) cryoconite albedo ($\alpha_c$-exp), broadband flux extinction coefficient of ice for the (g) direct component ($\kappa_d$-exp) and (h) diffuse component ($\kappa_f$-exp), (i) solar zenith angle ($\theta_z$-exp), and (j) zenith angle of the edge from the centre of the CH bottom ($\theta_c$-exp). Black lines in each figure indicate the control experiment ($Ctl$-exp). Note that lines for 15, 30 and 45° in the bottom right panel in the figure (j) are overlapped with the line for Control.

[Figure]

Figure 10: Daily mean temporal changes in direct and diffuse components of shortwave radiation reaching the cryoconite hole (CH) bottom in 2014. Blue and red lines indicate the direct ($R_{sdt}$+$R_{stdc}$) and diffuse ($R_{sfct}$+$R_{stfc}$) components in $R_S$-exp, respectively. Black line indicates both component of shortwave radiation ($R_{sdt}$+ $R_{sfct}$+ $R_{stdc}$+$R_{stfc}$) in Ctl-exp. Grey solid and dashed lines indicate the radiation components reaching the CH bottom from the hole mouth ($R_{sdt}$+ $R_{stdc}$) and transmitting through ice ($R_{sfct}$+ $R_{stfc}$) in Ctl-exp, respectively.

[Figure]

Figure 11: Sensitivity experiments of the cryoconite hole (CH) geometry on the direct and diffuse components of shortwave radiation reaching the CH bottom for (a, c) CH depth and (b, d) CH diameter. Dark blue and red bars in the figure indicate the direct and diffuse components of shortwave radiation, respectively. Light blue and red bars indicate the direct and diffuse components of shortwave radiation transmitted throughout ice, respectively. The vertical axis represents the fraction of direct and diffuse components against the incoming shortwave radiation at the ice surface (100 % at the ice surface). The ratios were derived from numerical simulations with different CH depth or CH diameter on day of the year (DOY) 172 in 2014. The meteorological conditions for the simulations were assumed to be those at 14:00 local time (a, b) and 1:00 local time (c, d) on the date used for Ctl-exp.

(2)Only melt at the hole centre is considered. This makes sense from the validation perspective as the measurements were collected at the centre. However, in Greenland or Antarctica as the sun goes round and round quite low in the sky it is plausible the hole centre is never directly illuminated but that the outer parts are illuminated directly for several hours. I think the two 'extremes' to quantify this would be the northern and southern edges of the hole bottom. Could some of the melt rates be repeated for these locations, with a simple adjustment to the geometry calculation? If it turns out the melt rate is

actually quite uniform across the bottom of the hole, that would itself be an interesting result as it would compare well with the generally flat hole bottoms and would support the model simply being applied to the centre and not the edges. This is related to the first point, since if the diffuse component is dominant then the melt rate should be quite uniform. On the other hand if the direct component has been underestimated then the melt rate could vary quite considerably across the hole bottom.

If the phenomenon pointed out by the reviewer occurs, the surface of CH bottom and the thickness of the cryoconite in the hole would be non-uniform. We guess that the non-uniformity causes positive feedback, resulting in the CH bottom being further non-uniform. Previous study suggests that topological conditions affect the CH geometry on steep north-sloping ice (Cook et al., 2018). However, the effect should be negligible because CHs observed in the studied glacier, where the slope is below 5°, were flat on the CH bottom.

To discuss the effects of the northern and southern edges of the CH bottom on the CH depth, we also conducted the sensitivity test of the CH depth to zenith angle of the edge from the centre of the CH bottom ($\theta_c$-exp). The experiment showed that $\theta_c$ hardly affects CH depth in the case of below 60 degrees (Figure 9 in the revised manuscript). The sensitivity test of the CH depth to initial depth ($D_0$-exp) suggests that the CH bottom is relatively accessible by shortwave radiation in the case of shallower depths, and vice versa, resulting in that the CH depth converge over time. Although the CH depth may be a temporary non-uniform in the northern and southern edges of the CH bottom, the CH depth is likely uniform again over time.

The above points have been described in Sections 5.2 and 5.4. Regarding the revised sentences, please see our response to your major comment (1). In Section 2.2, we have described that the model calculates the heat balance at centre of the CH bottom (Qc).


where $D_{t-1}$ is the CH depth at one time step before (m), and $\rho_w$ is the water density assumed equal to 1,000 kg m$^{-3}$. If the melt rate at the CH bottom is greater than that at the ice surface ($M_c > M_i$), the CH depth deepens, and vice versa. The initial depth $D_0$ at $t = 0$ in CryHo is a prescribed constant initial condition. Note that the heat balance at the CH bottom should vary on the position of the bottom such as the northern and southern edges. In this study, the heat balance at the center of the CH bottom was calculated for simplicity.

**Other points.**

Hole widening: I don't think this is specifically mentioned. However, the direct component is sensitive to hole diameter (because of shading) so it's certainly worth some discussion even if not included in this first version of CryHo. Later versions should probably attempt to track both the hole depth and the diameter. Is there some positive feedback? As the hole gets wider, more and more of the vertical wall gets illuminated, presumably causing further widening, and additionally a greater water surface area for turbulent heat transfer (noting the area increases as diameter^2 while the wall circumference for melting only increases linearly with diameter). As the hole widens, the bottom is also less shaded. Could that explain how cryoconite holes can sometimes grow quite large, eventually coalescing? Hole widening might mitigate collapse, to some extent, if it can help to increase depth.

Our sensitivity experiment ($\phi$-exp) suggests that the difference in hole diameter has little effect on hole depth. Furthermore, previous studies suggest that CH diameter is uncorrelated to CH depth (Gribbon, 1979; Cook, 2012). Cook (2012) formulated thermal energy directed to the hole walls. However, his observation in the Arctic region suggests that the CH diameter is not correlated with the CH depth but correlated with the thickness of cryoconite at the CH bottom. Therefore, the positive feedback suggested by the reviewer is unlikely to occur. On the other hand, Cook (2012) suggests that a portion of the absorbed radiation in the cryoconite at the CH bottom could be transferred laterally and then melt the CH wall. By incorporating the effect of the lateral heat as well as sensible heat in the water on the CH diameter into CryHo, temporal change in the CH diameter might be calculated.

The discussion regarding the feedback and the lateral heat has been described in Sections 5.2 and 5.4.

Figure 11b suggests no significant difference in the total shortwave radiation reaching the CH bottom among the different diameters, thereby supporting the $\phi$-exp result. Indeed, no significant correlation between the CH depth and diameter has been found (Gribbon, 1979; Cook, 2012). Since an increase in CH diameter cause more direct shortwave radiation reaching the CH bottom, positive feedback of the CH development is possible. However, such feedback is unlikely to occur because Figures 11b and 11d suggest that most of the diffuse component of the transmitted shortwave radiation reaches the CH bottom over 30 mm in CH diameter in the case of 10 mm depth. Furthermore, the observed CH depths and diameters significantly varied among the sites and years (Table 2), suggesting that CH depth is mainly controlled by factors other than the CH diameter.

Besides lingering water in CHs, the thickness of cryoconite at the CH bottom, which is not considered by CryHo, is also likely to be a key factor in determining the CH diameter and shape (Cook et al., 2010). This is likely because a portion of the absorbed radiation in the cryoconite at the CH bottom could be transferred laterally and then melt the CH wall (Cook, 2012).

L25: "wind speed". Technically should this be the turbulent heat transfer (which of course is a function of wind speed, but also other factors such as roughness, humidity, air temp etc).

The term has been modified to turbulent heat transfer. (Line 28)

L40-50: Our study from west Greenland also reported CH collapse and debris dispersal following warm/windy conditions (Chandler et al. 2005 TC; our Section 3.1). Also note we used four bare ice types rather than three (clean ice, dirty ice, CH, water).

Chandler et al. (2015) have been added (Line 47) because we think that Chandler et al. (2005) is a typo. The surface types have been changed to four types (Lines 53-55).

  CH collapse events have also been reported in Svalbard and Southwest Greenland (Hodson et al., 2007, 2008; Stibal et al., 2008; Irvine-Fynn et al., 2011; Chandler et al., 2015), while it has been indicated that a higher melt rate at the ice surface induces CH collapse (Hodson et al., 2007).

Topologically heterogeneous ice surfaces can be classified into four types: clean bare ice surfaces, dirty bare ice surfaces, surfaces with CHs, and meltwater streams (Irvine-Fynn and Edwards, 2014; Chandler et al., 2015; Holland et al., 2019; Tedstone et al., 2020).

Section 2: throughout the paper it would be better to write the equations in SI units, to avoid awkward conversion factors, even if you have used these other units in the model code for convenience.

We have unified the units to SI units throughout the whole of Section 2.

Eq 7: $Mc = t\_h * Q\_Mc / l\_M$ … the units don't balance here, I think there is a rho missing? Please could you double check all the equations for typos / consistent units (I haven't checked all of them).

Thank you for pointing out. The ice densities were missing in Eqs 5 and 7. The densities have been added to Eqs 5 and 7. Accordingly, the units of Q_Mi and Q_Mc are modified in the manuscript.

$$Q_{Mi} = \max[0, Q_i], \tag{5}$$

$$M_i = \frac{t_h Q_{Mi}}{l_M \rho_i}, \tag{}$$

$$Q_{Mc} = \max[0, Q_c], \tag{7}$$

$$M_c = \frac{t_h Q_{Mc}}{l_M \rho_i}, \tag{}$$

Eq 9-10: A diagram showing the ray paths of the four SW components would be handy here – maybe add a panel to Fig 1?

The diffuse component of shortwave radiation has been added to Figure 1 following your suggestion.

[Figure]

Figure 1: Concept of the cryoconite hole model (CryHo). Heat balances at the surface and cryoconite hole bottom are independently calculated (left). Red and orange arrows indicate direct and diffuse components of shortwave radiation, respectively. Cryoconite hole (CH) geometry, with depth ($D$) and diameter ($\phi$) being considered for distinguishing the direct component of shortwave radiation (right). Cryoconite thickness at the CH bottom is assumed to be zero in the model. The difference between the melt rate at the surface ($M_i$) and that at the CH bottom ($M_c$) changes the CH depth. The direct component of solar radiation can reach the CH bottom from the hole mouth if the solar zenith angle $\theta_z$ is smaller than the zenith angle of the CH edge $\theta_c$ (left, red solid arrow), while it is transmitted through the ice if the solar zenith angle is greater than the zenith angle of the CH edge (right, red dashed arrow). The diffuse component of downward shortwave radiation can reach the CH bottom regardless of $\theta_z$ (orange solid and dashed arrows).

Eq 18: Does it matter that the hole is full of water, in your LW calculations?

Although the model does not take into account the effect of water on the longwave radiation emitted from the CH wall, the effect on the simulation of the CH depth is probably little because the emissivity of the water is similar to that of the snow/ice surface (please see next our response to your comment).

L145: Why is the CH bottom temperature equal to the surface temperature? Shouldn't it simply be the melting point of ice? In cold conditions, under your assumption you would end up with the bottom of a water-filled hole cooling below 0C.

We have modified the explanation about the CH bottom temperature as suggested. In addition, Ti in Eqs (18) and (19) have been replaced with Tc.

Lines 154-159:

$$R_{Lw} = \cos^2 \theta_c \, \varepsilon \sigma T_c^{\,4}. \tag{18}$$

Here, we assume that the CH wall or bottom temperatures ($T_c$) are equal to the melting point. Since the CryHo does not calculate water level in the CH, the longwave radiation emitted from the CH wall is calculated using the emissivity of the snow/ice surface, which is similar to the emissivity of the water, in the Eq. (18). Regarding longwave radiation emitted from the CH bottom, the net longwave radiation ($R_{Lnc}$; W m$^{-2}$) can be summarised as follows:

$$R_{Lnc} = R_{Lc} + R_{Lw} - \varepsilon \sigma T_c^{\,4} = (1 - \cos^2 \theta_c) R_{Lni}. \tag{19}$$

Section 2.4: I would encourage the authors to consider in more detail the influence of partial shading as I noted earlier, and also the change in zenith angle as the radiation enters the water.

As our response to your major comments 1 and 2, we additionally conducted the sensitivity test to assess the sensitivity of the CH depth to the solar zenith angle ($\theta_z$-exp). Please see the major comments (1) and (2).

Also, does the transmittance through ice account for the low-density weathering crust? Maybe it doesn't matter for the deeper holes that are well below it, but something to consider in CryHo V2…

Previous study suggests that the density of the weathering crust is lower sometimes below 500 kg m$^{-3}$ (Müller and Keeler, 1969). Because the transmittance of the downward shortwave radiation would be larger in such low-density case than in ice density case (900 kg m$^{-3}$), the CHs under the weathering crust may also develop. We have described the future challenges in Section 5.2.

Reference:

Müller, F. and Keeler, M.: Errors in Short-Term Ablation Measurements on Melting Ice Surfaces, J. Glaciol., 8, 91–105, https://doi.org/10.3189/S0022143000020785, 1969.

Lines 421:426:

Further studies on the optical characteristics of ice are necessary, because there is little information on the broadband flux extinction coefficients. Furthermore, as reported previously, a porous ice layer known as weathering crust, commonly develops at the surface layer of glacial ice during summer (Irvine-Fynn and Edwards, 2014; Stevens et al., 2022). The density of the weathering crust is lower sometimes below 500 kg m$^{-3}$ (Müller and Keeler, 1969). Because the transmittance of the shortwave radiation would be larger in such a low-density case than in the ice density case assumed in this study (900 kg m$^{-3}$), the CHs under the weathering crust may also develop. In order to accurately simulate temporal change in CH depth, such weathering crust layer should be considered for the CH modeling in the future.

Section 3.3: I am quite untrusting of ablation measurements made using metal stakes, as they tend to melt into the ice unless they are installed deeply enough to be definitely remain well frozen at the bottom – could you comment further, or note it as a possible source of error (I think plastic tubes are better, but I acknowledge that opinions vary!)

In the observation, we buried the metal angle to the 1.5 m depth in the ice body for stabilization of the posture. The explanation has been added. (Lines 263-265)

Lines 263-265:

To collect in situ data that can be used for the evaluation of CryHo, the temporal changes in CH depth were observed using the monitoring device, which consisted of two time-lapse cameras as well as a plastic stick positioned in the centre of a CH, supported by metal angles buried to approximately 1.5 m depth in the ice body (Fig. 6a).

L296: A climate model can be used directly as a boundary condition rather than requiring any coupling, this is one of the great applications I can see for CryHo.

Thanks. We have a plan to do so.

L298: I didn't follow that sentence.

The sentence has been revised to clarify the content. (Lines 355-357)

Lines 355-357:

The sensitivity experiment regarding the shortwave radiation components ($R_S$-exp) suggests that

L385: Conclusion that diffuse component dominates, may need to be revised depending on what you find if accounting for refraction or partial shading as noted above.

As our response to your major comment (1), the air-water refraction and partial shading hardly affect CH depths in the studied glacier suggested by $\theta_z$-exp and $\theta_c$-exp. In addition, Rs-exp suggests that CHs tend to decay and develop in the case of that the direct and diffuse components are dominant, respectively. Furthermore, heat component analysis suggests that CH depth is governed by the balance between the intensity of the diffused component of downward shortwave radiation and the turbulent heat transfer. Based on these results, we have kept the conclusion described in the line 385.

Table 1: great if the symbols are listed alphabetically.

We have listed alphabetically each symbol shown in Table 1.

Fig 5: you have plotted a single model run based on some estimated parameters. Could you use several runs, covering a range of plausible parameters, and plot the range or stdev of resulting hole depths as a shaded band? It would help interpret the discrepancy between model and obs.

We additionally conducted model simulations using different parameters of ice surface albedo and plotted the results (Figures 7 and 8 in the revised manuscript). The parameters are based on the standard deviation of the ice surface reflectance shown in Table 2.

[Figure]

Figure 7: Temporal changes in cryoconite hole (CH) depth at (a) Site 1, (b) Site 2, (c) Site 3, and (d) Site 4 in 2014. The CH constants, ice surface albedo and air temperature correction are described in each panel. The black solid line and shading indicate model results using the average values and standard deviations of ice surface reflectance, which is assumed to be $\alpha_i$, shown in Table 2. The missing grey solid line in panel b denotes the CH collapse period.

Supplement: It would be great if this could be avoided completely. Figs S1-S4 could be combined as a multipart fig in the main text. The text on extinction coeffs could also be moved to the main text, it's an interesting part of the model. Fig S5 can join Fig 3.

The supplemental text on the extinction coefficient has been moved to Section 2.4. Figures S1 and S4 have been moved to the main text because these figures are especially important to parameterize the extinction coefficient (Figures 2 and 3 in the revised manuscript). Figure S5 was merged with Figure 3 (Figure 5 in the revised manuscript).

Fig S6 (move to main text), can the individual Rs components also be plotted separately?

Figure S6 has been moved to the main text (Figure 10 in the revised manuscript). The five plots cover all patterns of the radiation components (all, direct, diffuse, transmitted, and hole mouth components), so we don't think it is necessary to further refine the information. Instead of that, we have modified the labels in the figure legend to clarify each plot (you can see Figure 10 in our response to your major

comment (1)).

---

## Author Comment (AC2)

**Reply to Referee#2**

May 3, 2023
Dr. Yukihiko Onuma
Japan Aerospace Exploration Agency (JAXA)
E-mail: onuma.yukihiko@jaxa.jp

Dear Referee#2,

   We would appreciate a number of valuable comments very much. Please see enclosed our responses to all your comments as well as the revised marked-up manuscript entitled "Modelling the development and decay of cryoconite holes in Northwest Greenland" by Yukihiko Onuma et al. [Paper #egusphere-2023-54] submitted to the journal The Cryosphere. Our responses (**blue text**) to each of your comments (**black text**) were described on the following pages. We also described the revised sentences with the yellow marker following your suggestions.

Best regards,
Yukihiko Onuma and co-authors

**RC2: 'Comment on egusphere-2023-54', Anonymous Referee #2, 20 Mar 2023**

General statement:

The albedo of the Greenland Ice Sheet is of central importance to the surface energy budget. In the ablation area, the albedo is determined by whether debris is uniformly distributed or instead confined in cryoconite holes (CHs), so modeling the evolution of CHs is a worthwhile research project. The inputs to the model could be obtained from climate-model output. This paper could therefore be important, but in its current form it is difficult to read, so few readers will get through it.

The abstract could be improved by adding some key points, which are noted as they occur in the major comments below.

We would like to thank you very much for taking the time to review our manuscript. We are honored that you appreciate our project. According to suggestions from two reviewers, we have modified model code slightly and re-conducted numerical simulations including the sensitivity experiments. Accordingly, we have discussed about the results. The manuscript, figures and tables have been carefully revised to make those easier to understand the contents. The detailed our responses to your comments are as below.

Major comments:

(1) CHs develop because the albedo of cryoconite material (ac) is lower than the albedo of the surrounding bare ice (ai). It would therefore be good to explicitly examine the dependence of equilibrium CH depth on this difference (ai-ac), and add these results to the abstract.

Regarding the difference in albedo between ice surface and CH bottom (ai minus ac), you can confirm the difference from Figure 7(e, ai-exp). In the experiment, ac is constant of 0.1. The result indicates that CH tends to develop in the case of the greater difference between ai and ac as you know. Notably, the sensitivity tests showed that the CH depth does not equilibrate in any experiment cases at the studied glacier, where the meteorological conditions change before the depth reaches the equilibrium. It suggests that vertical dynamics of CHs mainly depends on not only the albedos but also meteorological conditions. According to your suggestion, we have added the point in Abstract and Section 5.2. (Lines 385-394).

Abstract:

**Abstract.** Cryoconite holes (CHs) are water-filled cylindrical holes with cryoconite (dark-coloured

sediment) deposited at their bottoms, forming on ablating ice surfaces of glaciers and ice sheets worldwide. Because the collapse of CHs may disperse cryoconite on the ice surface, thereby decreasing the ice surface albedo, accurate simulation of the temporal changes in CH depth is essential for understanding ice surface melt. We established a novel model that simulates the temporal changes in CH depth using heat budgets calculated independently at the ice surface and CH bottom based on hole-shape geometry. We evaluated the model with in situ observations of the CH depths on the Qaanaaq ice cap in Northwest Greenland during the 2012, 2014, and 2017 melt seasons. The model reproduced well the observed depth changes and timing of CH collapse. Although earlier models have shown that CH depth tends to be deeper when downward shortwave radiation is intense, our sensitivity tests suggest that deeper CH tends to form when the diffuse component of downward shortwave radiation is dominant, whereas CHs tend to be shallower when the direct component is dominant, although CH depth is unlikely to be correlated with CH diameter. In addition, the total heat flux to the CH bottom is dominated by shortwave radiation transmitted through ice rather than that directly from the CH mouths when the CH is deeper than 10 mm. Furthermore, the tests highlight that the difference in albedo between ice surface and CH bottom is a key factor for accurately reproducing the timing of CH collapse and that positive feedback of lower ice surface albedo induces CHs collapse and therefore causing further lowering of the albedo. Notably, the sensitivity tests showed that the CH depth does not equilibrate in any experiment cases at the studied glacier, where the meteorological conditions change before the depth reaches the equilibrium. Heat component analysis suggests that CH depth is governed by the balance between the intensity of the diffuse component of downward shortwave radiation and the turbulent heat transfer. Therefore, these meteorological conditions may be important factors contributing to the recent surface darkening of the Greenland ice sheet and other glaciers via the redistribution of CHs. Coupling the CH model proposed in this study with a climate model should improve our understanding of glacier-surface darkening.

Lines 385-394:

The sensitivity experiments regarding the albedos of the ice surface and CH bottom ($\alpha_i$-exp and $\alpha_c$-exp) suggest that the difference between ice surface ($\alpha_i$) and CH bottom albedos ($\alpha_c$) is an important factor for reproducing the CH dynamics, especially the timing of CH collapse. The CH depth increases with an increase in $\alpha_i$ owing to decreasing $M_i$, whereas it decreases with an increase in $\alpha_c$ owing to decreasing $M_c$ (Figs 9e and 9f). Notably, the sensitivity of the CH depth to $\alpha_i$ was greater than that to $\alpha_c$. This is probably because shortwave radiation at the ice surface was greater than that at the CH bottom. In addition, $\alpha_i$-exp suggests that a 0.1 decrease in $\alpha_i$ induces CH collapse one day earlier. Although $\alpha_c$ is known to be a key parameter for simulating the CH deepening rate (Podgorny and Grenfell, 1996), there is little information and discussions regarding $\alpha_i$. Since we used constant values of $\alpha_c$ and $\alpha_i$ in $\alpha_i$-exp and $\alpha_c$-exp, respectively, our results highlight for the first time the

importance of the difference between $\alpha_i$ and $\alpha_c$ in simulating vertical CH variations. CH tends to develop in the case of the greater albedo difference and vice versa.

(2) Equation 13. It is strange to compute the diffuse ratio under cloud from the net longwave at the surface, because the causality is backward: in reality the downward longwave is a consequence of cloud thickness.

As shown by van den Broeke et al. (2004), net longwave radiation at the surface becomes 0 under cloudy-sky conditions; whereas clear-sky condition can be recognized when net longwave radiation at the surface is negative and at a minimum for a given temperature. The reference has been added to the sentence of Eq. (13). (Line 130)

Reference:

van den Broeke, M., Reijmer, C., and van de Wal, R.: Surface radiation balance in Antarctica as measured with automatic weather stations, J. Geophys. Res., 109, D09103, doi:10.1029/2003JD004394, 2004.

(3) Eq. 15 (and other equations). These equations apply only to the center point of the CH. But parts of the bottom will still be in shadow for any nonzero solar zenith angle. There's no need to expand your calculations, but at least point out that you are ignoring this complication.

In response to Reviewer 1's comments about the zenith angles (major comments 1 and 2), we additionally conducted sensitivity tests to assess the sensitivity of the CH depth to solar zenith angle ($\theta_z$-exp) and zenith angle of the edge from the centre of the CH bottom ($\theta_c$-exp). Accordingly, we have updated Figure 7 (Figure 9 in the revised manuscript) and Table 3, and added discussion into Section 5.2 and 5.4.

Lines 111-115:

where $D_{t-1}$ is the CH depth at one time step before (m), and $\rho_w$ is the water density assumed equal to 1,000 kg m$^{-3}$. If the melt rate at the CH bottom is greater than that at the ice surface ($M_c > M_i$), the CH depth deepens, and vice versa. The initial depth $D_0$ at $t = 0$ in CryHo is a prescribed constant initial condition. Note that the heat balance at the CH bottom should vary on the position of the bottom such as the northern and southern edges. In this study, the heat balance at the center of the CH bottom was calculated for simplicity.

Lines 302-312:

We conducted sensitivity tests to assess the sensitivity of the CH depth to input data and model constants, such as air temperature ($T_a$-exp), radiation components ($R_S$-exp), initial depth ($D_0$-exp),

hole diameter ($\phi$-exp), albedo at the ice surface ($\alpha_i$-exp), albedo at the CH bottom ($\alpha_c$-exp), extinction coefficients of direct ($\kappa_d$-exp) and diffuse ($\kappa_f$-exp) radiation, solar zenith angle ($\theta_z$-exp), and zenith angle of the edge from the centre of the CH bottom ($\theta_c$-exp) (Table 3). Site-exp, i.e., Site 2 in 2014, was used as the control experiment for the sensitivity tests (Ctl-exp). The ranges of the changing parameters, which are summarized in Table 3, were determined based on field measurements (Table 2). The extinction coefficients for $\kappa_d$-exp and $\kappa_f$-exp were obtained from multiplying by factors of 0.25–4.00 the original values. The factor range was assumed by referring to the difference between the spectral flux extinction coefficient and absorption coefficient calculated from the imaginary refractive index of pure ice (Fig. 2). In $R_S$-exp, we assumed $r_{dif}$ of Eqs (9) and (10) to be 0 and 1 in Sd and Sf cases shown in Table 3, respectively. In $\theta_z$-exp and $\theta_c$-exp, $\theta_z$ and $\theta_c$ calculated in the model were replaced with the values shown in Table 3, respectively, in order to quantify effects of the zenith angles on the CH depth.

Lines 427-439:

The sensitivity experiments regarding the zenith angle ($\theta_z$-exp and $\theta_c$-exp) suggest that differences in the zenith angles have little influence on the CH depth, except for the case of that the downward shortwave radiation always reaches the CH bottom from the hole mouth ($\theta_z = 0°$). $\theta_z$-exp showed that the CH depth with a higher $\theta_z$ was shallower, owing to a decrease in $M_c$ (Fig. 9i). In contrast, $\theta_c$-exp showed that the CH depth with a lower $\theta_c$ was smaller (Fig. 9j). Notably, the experiments suggest that $\theta_z$ and $\theta_c$ hardly affect the CH depth in the case of over 15° and below 60°, respectively. Snell's law states that direct component of incident radiation is refracted through the air-water surface. The refraction angle is smaller approximately 20° than the incident angle $\theta_z$ by the law, therefore the direct component of the downward shortwave radiation more easily reaches the CH bottom from the hole mouth. However, such refraction is unlikely to affect the CH depth. Although the CryHo calculates $M_c$ at centre of the CH bottom using $\theta_c$, $M_c$ may differ at the northern and southern edges of the CH bottom because the zenith angles of the edges of the CH bottom would differ from $\theta_c$. $\theta_c$-exp suggests that the CH depth is a temporary non-uniform in the northern and southern edges of the CH bottom. However, the CH depth is likely uniform again over time according to $D_0$-exp. Indeed, the simulated CH depth using the different $\theta_c$ converged within approximately two weeks (Fig. 9j). In addition, CHs observed in the studied glacier were flat on the CH bottom.

Lines 471-482:

Our model does not include the effect of water lingering in CHs on the heat balance at the CH bottom because a quantitative understanding of the mechanism of convective heat transport or the buffering effect in the lingering water is insufficient. Such lingering water in CHs may affect the heat exchange between the atmosphere and CH bottom. Heat exchange should not be negligible in the case of large

water surfaces in CHs. Although water level in CHs is not estimated in the model, the refraction through air-water surface in CHs unlikely to affect CH depth in the studied glacier as discussed in $\theta_z$-exp. However, the refraction might contribute to CH development in the lower latitude regions such as Asia, where the solar zenith angle is significantly smaller than that in polar region. In addition to the refraction, reflectance at the water surface would reduce amount of shortwave radiation reaching the CH bottom. To simulate CH depths globally, such an effect may have better been incorporated into CryHo. Besides lingering water in CHs, the thickness of cryoconite at the CH bottom, which is not considered by CryHo, is also likely to be a key factor in determining the CH diameter and shape (Cook et al., 2010). This is likely because a portion of the absorbed radiation in the cryoconite at the CH bottom could be transferred laterally and then melt the CH wall (Cook, 2012).

[Figure]

Figure 9: Sensitivity experiments of the temporal changes in cryoconite hole (CH) depth to model parameters and meteorological conditions at Site 2 in 2014. (a) Air temperature ($T_a$-exp), (b) shortwave radiation ($R_S$-exp), (c) initial CH depth ($D_0$-exp), (d) CH diameter ($\phi$-exp), (e) ice surface albedo ($\alpha_i$-exp), (f) cryoconite albedo ($\alpha_c$-exp), broadband flux extinction coefficient of ice for the (g) direct component ($\kappa_d$-exp) and (h) diffuse component ($\kappa_f$-exp), (i) solar zenith angle ($\theta_z$-exp), and (j) zenith angle of the edge from the centre of the CH bottom ($\theta_c$-exp). Black lines in each figure indicate the control experiment ($Ctl$-exp). Note that lines for 15, 30 and 45° in the bottom right panel in the figure (j) are overlapped with the line for Control.

(4) Eq. 20 and elsewhere.    The factor of 1000 is distracting, and it is not necessary; the user just needs to keep track of the units of k and D.

We have unified the unit of D to "m" throughout the whole of Section 2.

(5) Equation 21 for RStfc is wrong, because the path length of diffuse transmission into ice is taken to be just the vertical distance (as pointed out on lines 155-156). Instead you need to use the diffusivity factor (for example Liou 1980 p 97 eq 4.26): The effective path length for diffuse flux is the product of the vertical distance and the average secant for diffuse radiation, which varies with depth, but is often taken to be the secant of 53 degrees, i.e. 1.66.

Reference:

Liou, K.N., 1980: An Introduction to Atmospheric Radiation. Academic Press.

In Eq. (21), we parameterized the broadband flux extinction coefficient of bare ice based on spectral flux extinction coefficients experimentally determined by Cooper et al. (2021) for ice on the GrIS. Cooper et al. (2021) measured the vertical profile of the spectral flux extinction coefficients for the diffuse component of shortwave radiation, by which we parameterized the value of $\kappa_f$ as a function of ice thickness $x$. Therefore, it is not necessary to multiply the depth by 1.66 for the vertical depth of bare ice in Eq. (21).

 On the other hand, we noticed from your comment that the approximated value 1.66 should be used to obtain $\kappa_d$ from $\kappa_f$. We do not know the exact values of $\kappa_d$. Considering a path length $x$, transmittance of the direct component of downward shortwave radiation ($T_d$) is described as follows:

$T_d = \exp(-\kappa_d {}^* x)$

Since the effective optical path length for the diffuse component can be approximated as $1.66x$, the below relationship is derived.

$\kappa_f x \sim \kappa_d 1.66x$

Therefore, $T_d$ is described as follows:

$T_d = \exp(-\kappa_f x / 1.66)$

Hence, we have revised the relationship between $\kappa_d$ and $\kappa_f$ as:

$\kappa_d = \kappa_f / 1.66$, (Equation (29) in the revised manuscript).

Accordingly, all numerical simulations have been re-conducted. Based on the results, we have revised the discussions in the manuscript. In addition, supplemental text for parameterization of the extinction coefficient has been moved to the main text according to Reviewer 1's suggestion.

Lines 232-234:

The coefficient $r_d$ was assumed to be 1/1.66. The value 1.66 is used as an approximation value of the effective optical path length when the transmittance of diffuse component of shortwave radiation is obtained from that of direct component of shortwave radiation (Liou, 1980). Based on the assumption, $\kappa_d$ was obtained by dividing $\kappa_f$ by 1.66.

(6). Lines 320-324 point out that the CH depth is uncorrelated with CH diameter. This is an important result which should be included in the abstract.

The result has been added to Abstract. (Line 21)

(7) Lines 334-336. The positive feedback of low ice-albedo (ai), causing CHs to collapse and therefore causing further lowering of ai, is important and should be included in the abstract.

The positive feedback has been described in Abstract. (Line 25)

(8) Line 332: "Our results highlight for the first time the importance of both ai and ac". The key variable is probably neither ai nor ac, but rather their difference (ai-ac). It would be good to add a figure plotting equilibrium depth versus (ai-ac) for the standard values of other inputs. This is related to comment (1).

As our response to comment (1), we have emphasized the importance of the difference in Abstract and Section 5.2. (please see our response to your major comment (1)).

(9) I cannot make sense of lines 341-348; they need to be rewritten. For example, I don't understand "the direct component of shortwave radiation is transmitted throughout the ice rather than the diffuse component." Also, how can "diffuse" be "direct", as in this statement: "The CH bottom is directly accessible by a part of the diffuse component."

We have carefully revised the text to clarify the contents. (Lines 403-410)

Lines 403-410:

The sensitivity experiments regarding the broadband extinction coefficients of shortwave radiation transmitted throughout ice for the direct and diffuse components ($\kappa_d$-exp and $\kappa_f$-exp) suggest that $\kappa_f$ is more effective at the CH depth than $\kappa_d$. Both experiments showed that the CH depth with a higher coefficient was shallower, owing to a decrease in $M_c$ (Figs 9g and 9h); however, there was a significant difference in the sensitivity to the coefficients. One of the reasons is probably that $\kappa_d$ is lower than $\kappa_f$, as in Eq. (29). Figure 10 showed that the diffuse component of shortwave radiation reaching the CH bottom was greater than the direct component of that even though $\kappa_f$ is higher than $\kappa_d$, suggesting that the diffuse component of shortwave radiation reaches the CH bottom more easily than the direct component. Indeed, Figure 11 indicates that the reaching fraction of direct component

of shortwave radiation transmitted throughout the ice to the CH bottom decreases with $\theta_z$.

(10) Lines 370-371. The observation that CHs decay under overcast cloud is probably not because of the diffuse nature of the incident radiation.   Under a cloud, the total downward shortwave is dramatically reduced, which means that turbulent fluxes become a larger fraction of the total energy budget, leading to CH decay.

The sentence has been modified as you suggested. (Lines 450-455)

Lines 450-455:

Takeuchi et al. (2018) suggested that CHs tend to be shallower under both cloudy and windy conditions. Our analyses also suggest that windy conditions are important meteorological conditions governing the CH decay (Figure 13a). Because the total downward shortwave is reduced through clouds although the diffuse component of downward shortwave radiation generally increases under cloudy condition, turbulent fluxes are likely to be dominant to the total energy budget, resulting in that the ice surface melts faster than the CH bottom. This is probably the reason why the CH collapse events were observed under cloudy conditions.

(11) Figure 1b. Only part of the CH bottom is shaded; the rest is sunlit.

Diffuse component of downward shortwave radiation was missing in Figure 1. So, we have added the lines for diffuse component of downward shortwave radiation into the figure. (Figure 1)

[Figure]

Figure 1: Concept of the cryoconite hole model (CryHo). Heat balances at the surface and cryoconite hole bottom are independently calculated (left). Red and orange arrows indicate direct and diffuse components of shortwave radiation, respectively. Cryoconite hole (CH) geometry, with depth ($\boldsymbol{D}$) and

diameter (**φ**) being considered for distinguishing the direct component of shortwave radiation (right). Cryoconite thickness at the CH bottom is assumed to be zero in the model. The difference between the melt rate at the surface ($M_i$) and that at the CH bottom ($M_c$) changes the CH depth. The direct component of solar radiation can reach the CH bottom from the hole mouth if the solar zenith angle $\theta_z$ is smaller than the zenith angle of the CH edge $\theta_c$ (left, red solid arrow), while it is transmitted through the ice if the solar zenith angle is greater than the zenith angle of the CH edge (right, red dashed arrow). The diffuse component of downward shortwave radiation can reach the CH bottom regardless of $\theta_z$ (orange solid and dashed arrows).

(12) Figure 3 is completely mysterious to me, so it needs to be redrawn. The long dark bars (apparently meaning LH?) extend on both sides of zero; what does that mean? The long dark bars are shaded where they are above zero; what does that mean? Some of the long dark bars have a short gap just below the zero line, then they continue as a tiny black box below the gap; what does that mean? What are the short dark bars at the top of some of the bars? Which bars are net radiation?

Figure 3 has been redrawn to clarify each bar. In addition, the explanation of the positive sign of each energy flux in the figure has been added to the figure caption. The positive sign of each energy flux means a downward component. Figure S5 has been merged with Figure 3 (Figure 5 in the revised manuscript) following Reviewer 1's comment.

[Figure]

Figure 5: (a) Hourly air and surface temperatures ($T_a$ and $T_s$ in the upper panels, respectively) and daily surface energy balance (lower panels) at the Sigma-B site (Site 5 in this study) on the Qaanaaq ice cap during the 2012, 2014, and 2017 summer seasons from left to right, respectively. (b) Hourly air temperature (upper panels) and daily surface energy balance (lower panels) at Site 2. $T_s$ shown in the upper panels in the figure (a) is calculated from the downward and upward longwave radiations. $T_a$ shown in the upper panels in the figure (b) is corrected from $T_a$ in the figure (a) and an observed lapse rate. Daily surface energy balance is calculated with CryHo. The positive sign of each energy flux means a downward component.

(13) The paper is difficult to read, partly because the reader needs to keep track of the non-intuitive subscripts on the variables. Unfortunately, I don't have useful suggestions on what to do about this.

According to the comment from Reviewer 1, we have listed alphabetically each symbol shown in Table 1 in order to easily trace each variable and constant used in this study. In addition, the supplemental text has been moved to the main text.

Minor comments:

Line 49. Is "bare ice" uncontaminated, or does it contain distributed cryoconite material?
Cryoconite particles are distributed on bare ice. The ice surface is known as dirty bare ice. Reviewer 1 suggested that bare ice should be divided into two types: dirty bare ice and clean bare ice, so the sentence has been modified (Lines 53-55).

Lines 53-55:
Topologically heterogeneous ice surfaces can be classified into four types: clean bare ice surfaces, dirty bare ice surfaces, surfaces with CHs, and meltwater streams (Irvine-Fynn and Edwards, 2014; Chandler et al., 2015; Holland et al., 2019; Tedstone et al., 2020).

Line 76. "latent heat flux". Point out that HLi is restricted to the latent heat of evaporation; it does not include the latent heat of melting.
The explanation has been added. (Lines 80-85)

Lines 80-85:
where $\alpha_i$ is the ice surface albedo (dimensionless), $R_S$ is the downward shortwave radiation (W m$^{-2}$), $R_{Lni}$ is the net longwave radiation (W m$^{-2}$), $H_{Si}$ is the sensible heat flux (W m$^{-2}$), $H_{Li}$ is the latent heat flux (W m$^{-2}$), $\varepsilon$ is the emissivity of the snow/ice surface, which is assumed to be 1.0 (dimensionless), $R_L$ is the downward longwave radiation (W m$^{-2}$), $\sigma$ is the Stefan–Boltzmann constant (5.67 × 10$^{-8}$ W m$^{-2}$ K$^{-4}$), and $T_i$ is the surface temperature (K). Subscript $i$ used in the variables refers to the ice surface. All the downward components have positive signs. Note that $H_{Li}$ is restricted to the latent heat of evaporation; it does not include the latent heat of melting.

Lines 79-80. "Heat conduction from the glacier ice . . . assumed to be negligible." This is valid if Ta is never colder than Ts. Figure 3 shows that this condition holds during the summer: whenever Ta is negative, Ts likewise is negative and approximately equal to Ta. But in spring and autumn they might differ. This "Ts" in Figure 3 is probably what is called "Ti" in the text, neither of which is defined in Table 1.
Thank you for the comment. Ts was derived from the automatic weather station, so Ts differs from Ti, which is calculated in the CryHo. Ts has been added into Table 1.

Line 85. Give a reference for the value of bulk coefficient.

The coefficient is based on the below reference. The reference has been added to the sentence. (Line 90)

Reference:

Kondo, J.: Meteorology of water environment, Asakura Publishing, Tokyo, Japan, 1994.

Line 96 Eq 7 for Mc.   The units don't match.   LHS is mm/hour, but RHS is kg m^-2 hr^-1. The missing factor is density, kg m^-3.

Thank you for pointing out. The ice densities were missing in Eqs 5 and 7. The densities have been added to the Eqs 5 and 7. Accordingly, the units of Mi and Mc are modified in the manuscript.

$$Q_{Mi} = \max[0, Q_i],\qquad(5)$$

$$M_i = \frac{t_h Q_{Mi}}{l_M \rho_i},$$

$$Q_{Mc} = \max[0, Q_c],\qquad(7)$$

$$M_c = \frac{t_h Q_{Mc}}{l_M \rho_i},$$

Line 112. Change "distinguish" to "separate".

The word has been modified.

Line 121.   Niwano et al. 2015 is missing from the reference list.

The reference has been added to the list.

Line 155. Change "pass length" to "path length".

The word has been modified. (Line 167)

Line 171. RSfd and RSfs are undefined; they do not appear in Table 1.

These terms were typos. Those terms have been modified to Rsd and Rsf, respectively. (Line 225)

Line 177. Eq (25). To say that rd=1.0 is equivalent to assuming that the incident radiation becomes rapidly diffused in the topmost millimeters of the ice, which is probably true.

As our response to your major comment (5), rd has been assumed to be 1/1.66 for the CryHo.

Line 236. "Lapse rate" is the rate of decrease of temperature with height. If temperature decreases with height, the lapse rate is therefore positive. So remove the minus-sign, unless you mean a temperature-inversion.

The minus-sign has been removed. (Line 291)

Line 428 says that KF designed the study, but then the next sentence says that it was instead YO, NT, and TA who designed the study.

The sentence has been modified to the below. (Lines 515-517)

Lines 515-516:

KF designed the study. YO, NT, and TA designed the ==field observations==. KF developed the model with the support of MN and TA.

Table 1. Does C have units?

The C does not have a unit (non-dimension).

Figure 3 caption line 601, "Ts". In Table 1 you instead use Ti.

Ts means surface temperature derived from the automatic weather station. The temperature is not used for the model simulations. Ts has been added to Table 1.

Figure 5. Depth and diameter have unnecessary zeros after the decimal points. For example, change 39.0 to 39.

The decimal points have been removed from Figures 5 and 6 (Figures 7 and 8 in the revised manuscript).

Figure 7a,f,g,h. Reverse the order of the legends to correspond with the order of curves. For example, in 7a the red curve is on top, so the legend should have the red legend (+3) on top.

The figure panels have been modified as you suggested (Figure 9 in the revised manuscript).

Figure S3. The horizontal azis is labeled theta-0. In the text it is theta-z.

The theta-0 has been changed to the theta-z (Figure S2 in the revised supplemental text).

Figure S6. Three of the plots are labeled "control"; what does that mean?

These plots have different components from each other. We have modified the labels in the figure legend. In addition, Figure S6 has been moved to the main text (Figure 10 in the revised manuscript).

[Figure]

Figure 10: Daily mean temporal changes in direct and diffuse components of shortwave radiation reaching the cryoconite hole (CH) bottom in 2014. Blue and red lines indicate the direct ($R_{sdt}+R_{stdc}$) and diffuse ($R_{sfct}+R_{stfc}$) components in $\boldsymbol{R_S}$-exp, respectively. Black line indicates both component of shortwave radiation ($R_{sdt}+R_{sfct}+R_{stdc}+R_{stfc}$) in Ctl-exp. Grey solid and dashed lines indicate the radiation components reaching the CH bottom from the hole mouth ($R_{sdt}+R_{stdc}$) and transmitting through ice ($R_{sfct}+R_{stfc}$) in Ctl-exp, respectively.

---

## Author Response (AR1)

**Reply to Dr. Benjamin Smith (Editor)**

May 30, 2023 Dr. Yukihiko Onuma Japan Aerospace Exploration Agency (JAXA) E-mail: onuma.yukihiko@jaxa.jp

Dear Dr. Smith,

Thank you very much for handling our manuscript. Please see enclosed our responses to your and all reviewer's comments as well as the revised marked-up manuscript entitled "Modelling the development and decay of cryoconite holes in Northwest Greenland" by Yukihiko Onuma et al. [Paper #egusphere-2023-54] submitted to the journal The Cryosphere. Our responses (**blue text**) to each of your and reviewer's comments (**black text**) were described on the following pages. We also described the revised sentences with the yellow marker following your suggestions. Please note that our responses to the comments from two reviewers are generally same as those posted on the Discussion board in TCD, but the yellow marked sentences have been updated.

Best regards, Yukihiko Onuma and co-authors

**10 May 2023**

**Editor decision: Publish subject to minor revisions (review by editor) by Benjamin Smith**

**Public justification (visible to the public if the article is accepted and published):**

Thanks to the authors for their thorough response to the referees. It seems to me that the concerns of the referees are handled well, and that the authors should move towards submitting a revised manuscript. I will take care of the final edits.

Additional private note (visible to authors and reviewers only):

My two recommendations for revising and submitting the manuscript are:

(1) When explaining the choice of calculating the heat balance at the bottom of the CH based on conditions at the center of the hole, the authors propose to add this sentence in their revision to lines 111-115:

"Note that the heat balance at the CH bottom should vary on the position of the bottom such as the northern and southern edges. In this study, the heat balance at the center of the CH bottom was calculated for simplicity."

It would be reasonable at this point to provide some of the justification that was described in the response to reviewers, or at least to point ahead to the modeling experiments where it was shown that the variation in illumination is likely unimportant. For example, this sentence might say"

"Note that the heat balance at the CH bottom should vary across the width of the hole, particularly between the northern and southern edges. We demonstrate in our modeling experiments that this variation is not a dominant factor in the development of CH, as is confirmed by field observations that the bottoms of CH are generally flat. Therefor, for simplicity, we represent the heat balance based on the center of the CH bottom"

Second, I noticed that the quality of the English is high throughout the manuscript, but somewhat less so in the highlighted text presented in the response to the referees. It looks like the ideas are expressed correctly, but the way that they are written is a little bit hard to follow. Please make sure that this text has been edited before resubmitting the manuscript.

We would like to thank you very much for taking the time to check our manuscript. We glad that you accept our reply to the reviewers. We have carefully checked the revised manuscript. We requested English proofreading to improve the quality of the revised manuscript, so the manuscript has been checked by a native English speaker. Regarding the lines 111-115, a part of the recommended sentences has been moved to Section 4 (Lines 307-312) to explain why we did conduct the sensitivity test of hole edges to the CH depth ( $\theta_c$ -exp).

**Lines 110-112:**

If the melt rate at the CH bottom is greater than that at the ice surface  $(M_c > M_i)$ , the CH depth deepens, and vice versa. The initial depth  $D_0$  at t = 0 in CryHo is a prescribed constant initial condition. For simplicity, we assume the heat balance at the centre of the CH bottom can represent that of the entire bottom.

**Lines 307-312:**

In  $R_s$ -exp, we assumed  $r_{dif}$  of Eqs (9) and (10) to be 0 and 1 in Sd and Sf cases shown in Table 3, respectively. Because we did not consider the light refraction at the air-water surface in the CH, we evaluated the refraction effect via sensitivity analysis ( $\theta_z$ -exp) using various incident angles ( $\theta_z$ ). Although the heat balance at the CH bottom should vary across the width of the hole, particularly between the northern and southern edges, we assumed heat balance only at the centre of the CH bottom for the model simulation. The effect of different edges is evaluated by changing  $\theta_c$  ( $\theta_c$ -exp). In  $\theta_z$ -exp and  $\theta_c$ -exp,  $\theta_z$  and  $\theta_c$  calculated in the model were replaced with the values shown in Table 3, respectively.

**RC1: 'Comment on egusphere-2023-54 (Yukihiko Onuma et al.)', David Chandler, 16 Mar 2023**

I'd like to thank the authors for their efforts developing this new model for cryoconite hole depth, which is well presented along with useful sensitivity experiments and some validation. As the authors point out, changes in cryoconite hole dynamics can influence ice surface albedo – so this is an important topic, given that SMB is one of the key controls on Greenland's sea-level contribution. I imagine this model could easily be driven by either AWS data or climate model output, making it a useful tool for investigating how cryoconite holes could influence albedo under climate warming anywhere in Greenland or indeed Antarctica given some basic observations of typical hole dimensions (which are already available for many places). Other applications would include supraglacial hydrology (changes in water storage) and ice surface microbial processes.

The model calculates changes in hole depth by considering energy balance at the centre of the hole. Validation with some field observations yields an encouraging match overall, with some discrepancies as we would expect.

There are two important aspects which I think need to be considered further before publication, given the application of this model in regions with generally large zenith angles. On that basis I have ticked the major revisions box, but I'm hoping it's not a lot of work to implement these changes. Elsewhere there are some minor points requiring additional clarification, and the manuscript needs language editing by a native English speaker as there are numerous grammatical errors and a few sentences which are a little hard to follow. Apart from the language itself, the paper is clear and easy to follow.

I'm not very up to date with the relevant literature so I just reviewed this study on its own merits and not in relation to other recent work.

We would like to thank you very much for taking the time to review our manuscript. Since the cryoconite hole model (CryHo) could be driven by climate model output as you mentioned, the model has a potential to spatio-temporally evaluate albedo reduction caused by collapses of CHs in the Greenland Ice Sheet under climate change. According to the suggestions from you and another reviewer, we have modified model code slightly, re-conducted numerical simulations including the sensitivity experiments, and revised the manuscript. The detailed our responses to your comments are as below.

**Main points**

(1) Refraction is not considered when the direct SW component passes from air to water. I wonder if

that would change your conclusion that the diffuse component dominates over the direct component. If the zenith angle (in the air) is theta\_a, and the refractive indices of air and water are  $n_a = 1$  and  $n_w = 1.33$ , then the zenith angle in the water (theta\_w) would be estimated from Snell's law, i.e.,

 $n_a * sin(theta_a) = n_w * sin(theta_w).$

This is worth considering, since your range of zenith angles in air (noted as 56 to 85deg: Line 305) would become 38 to 48deg in water, so it's much more likely that the direct SW can reach the hole centre. You might also want to consider reflection by the water surface.

Refraction along the transmitted (air-ice-water) pathway would also be worth considering but I imagine would be harder to implement.

I think it would be quite easy to adjust the model to account at least for this air-water refraction and hopefully not a lot of work to re-run the plotting scripts so we can see if this is important or not.

Thank you for your constructive comments. As you pointed out, the air-water refraction of light might increase the amount of solar radiation reaching the CH bottom even when the solar zenith angle is larger, while the opposite effect would occur when the reflection of light at the water surface reduces the amount of that reaching the CH bottom. Since these two effects depend on water depth, it is difficult to incorporate the effect of the refraction and reflectance into the model, which does not simulate water level in CH. However, we additionally conducted a sensitivity test to the solar zenith angle ( $\theta_z$ -exp) to discuss the effect of the zenith angle on the CH depth. The experiment showed that the solar zenith angle hardly affects CH depth in cases over 15° (Figure 9). This is probably due to that the direct component of downward shortwave radiation hardly reaches the CH bottom from the hole mouth in such cases. In the studied glacier, the solar zenith angle generally ranges from 56 to 85°, suggesting that the contribution of light refraction through air-water boundary to CH depth is insufficient. We add the result and discussion in Section 5.2 as well as the explanation about the sensitivity test in Section 4. The refraction may have better been considered to simulate CH depth globally because the solar zenith angle is sometimes below 20° in low latitudes such as Asia. This point has been raised in Section 5.4 as future challenge.

Regarding our conclusion that CHs tend to decay and develop in the case of that the direct and diffuse components are dominant, we have discussed the sensitivity of the CH depth to the shortwave radiation components in the paragraph for  $R_s$ -exp. Figures 9b and 10 in the revised manuscript showed that the CH tends to develop when the diffuse component is dominant. In the studied glacier, there is no significant difference between the direct and diffuse components reaching the CH bottom even at the time of meridian transit in the summer solstice (Figures 11a and 11c in the revised

manuscript). The result suggests that the diffuse component relatively reaches CH bottom more than the direct component in the studied glacier. This may be one of the reasons why the diffuse component contributes to CH development rather than the direct component in  $R_s$ -exp. We have added the result of  $R_{sc}D$ -exp and  $R_{sc}\phi$ -exp in the higher  $\theta_z$  case (Figure 11) and the discussion in Section 5.2.

**Lines 299-318:**

We conducted sensitivity tests to assess the sensitivity of the CH depth to input data and model constants, such as air temperature ( $T_a$ -exp), radiation components ( $R_s$ -exp), initial depth ( $D_0$ -exp), hole diameter ( $\phi$ -exp), albedo at the ice surface ( $\alpha_i$ -exp), albedo at the CH bottom ( $\alpha_c$ -exp), extinction coefficients of direct ( $\kappa_d$ -exp) and diffuse ( $\kappa_f$ -exp) radiation, solar zenith angle ( $\theta_z$ -exp), and zenith angle of the edge from the centre of the CH bottom ( $\theta_c$ -exp) (Table 3). Site-exp, i.e., Site 2 in 2014, was used as the control experiment for the sensitivity tests (Ctl-exp). The ranges of the changing parameters, which are summarised in Table 3, were determined based on field measurements (Table 2). The extinction coefficients for  $\kappa_d$ -exp and  $\kappa_f$ -exp were obtained from multiplying by factors of 0.25–4.00 the original values. The factor range was assumed by referring to the difference between the spectral flux extinction coefficient and absorption coefficient calculated from the imaginary refractive index of pure ice (Fig. 2). In  $R_s$ -exp, we assumed  $r_{dif}$  of Eqs (9) and (10) to be 0 and 1 in Sd and Sf cases shown in Table 3, respectively. Because we did not consider the light refraction at the air-water surface in the CH, we evaluated the refraction effect via sensitivity analysis ( $\theta_z$ -exp) using various incident angles ( $\theta_z$ ). Although the heat balance at the CH bottom should vary across the width of the hole, particularly between the northern and southern edges, we assumed heat balance only at the centre of the CH bottom for the model simulation. The effect of different edges is evaluated by changing  $\theta_c$  ( $\theta_c$ -exp). In  $\theta_z$ -exp and  $\theta_c$ -exp,  $\theta_z$  and  $\theta_c$  calculated in the model were replaced with the values shown in Table 3, respectively.

[revised manuscript text omitted]

---

## Author Response (AR2)

**Author's response**

**20 June 2023**

**Editor decision: Publish subject to minor revisions (review by editor) by Benjamin Smith**

**Public justification (visible to the public if the article is accepted and published):**

The revised manuscript looks good. I have just a few revisions that need to go in for the last two sections. Once these are cleaned up, we should be able to wrap the manuscript up right away.

Best

Ben

We would like to thank you very much again. We really appreciate your thorough check of our revised manuscript. Please see our responses below.

Revisions:

481: the refraction -> such refraction

483-84: "may have been" -> "may need to be"

The sentences have been revised (Lines 481 and 483-484).

490: ", in the case of a cylindrical CH on a rather flat surface with a diameter of 0.10 m," -- is this intended to be a limiting case? If so, then the better statement would be "even for relatively large CH on flat surfaces with diameters up to around 0.1 m". If I have misunderstood this, please add some words to clarify the intent of this phrase.

Yes, the sentence means a limiting case. The sentence has been revised as suggested (Line 490).

493: specify "The CH diameter range considered in this study"

The range is based on that assumed for $\phi$-exp. The information has been added (Lines 492-493).

504 "relatively accessible"-- should this be "more accessible"? Relatively does not indicate more or less. Or is the point that both diffuse and direct radiation can influence CH development?

"more accessible" is correct for the sentence. The word has been modified (Line 504).

505- should be "both components"

I missed that. The word has been modified (Line 505).